# Temporal multiplexing of perception and memory codes in IT cortex

Liang She[1✉], Marcus K. Benna[2,3], Yuelin Shi[1], Stefano Fusi[2] & Doris Y. Tsao[1,4,5✉]

A central assumption of neuroscience is that long-term memories are represented by the same brain areas that encode sensory stimuli[1]. Neurons in inferotemporal (IT) cortex represent the sensory percept of visual objects using a distributed axis code[2–4]. Whether and how the same IT neural population represents the long-term memory of visual objects remains unclear. Here we examined how familiar faces are encoded in the IT anterior medial face patch (AM), perirhinal face patch (PR) and temporal pole face patch (TP). In AM and PR we observed that the encoding axis for familiar faces is rotated relative to that for unfamiliar faces at long latency; in TP this memory-related rotation was much weaker. Contrary to previous claims, the relative response magnitude to familiar versus unfamiliar faces was not a stable indicator of familiarity in any patch[5–11]. The mechanism underlying the memory-related axis change is likely intrinsic to IT cortex, because inactivation of PR did not affect axis change dynamics in AM. Overall, our results suggest that memories of familiar faces are represented in AM and perirhinal cortex by a distinct long-latency code, explaining how the same cell population can encode both the percept and memory of faces.

Our experience of the world is profoundly shaped by memory. Whether we are shopping for a list of items at the grocery store or talking to friends at a social gathering, our actions depend critically on remembering a large number of visual objects. Multiple studies have explored the molecular[12,13] and cellular[14,15] basis for memory, but the network-level code remains elusive. How is a familiar song, place or face encoded by the activity of neurons?

Recent work on the sensory code for visual object identity in the inferotemporal (IT) cortex suggests that objects are encoded as points in a continuous, low-dimensional object space, with single IT neurons linearly projecting objects onto specific preferred axes[2–4] (Fig. 1a, left). These axes are defined by weightings of a small set of independent parameters spanning the object space. This coding scheme (also referred to as linear mixed selectivity[16,17], and related to disentangled representations in machine learning[18]) is efficient, allowing a huge number of different objects to be represented by a small number of neurons. Indeed, the axis code carried by macaque face patches allows detailed reconstruction of random realistic faces using activity from only a few hundred neurons[3].

Here we set out to leverage recent insight into the detailed sensory code for facial identity in IT cortex[3] to explore the population code for face memories. A long-standing assumption of neuroscience is that long-term memories are stored by the same cortical populations that encode sensory stimuli[1]. This suggests that the same neurons that carry a continuous, axis-based, object-coding scheme should also support tagging of a discrete set of remembered objects as familiar. However, schemes for representing discrete familiar items often invoke attractors[19,20] that would lead to breakdowns in continuous representation (Fig. 1a, right). This raises a key question: does familiarity alter the IT axis code for facial identity? We surmised that discovering the answer might uncover the neural code for face memory.

Previous studies have generally found decreased and sparsened responses to familiar stimuli in IT and perirhinal cortex and have proposed that this decrease, or 'repetition suppression', is the neural correlate of object memory[5–11]. However, these studies were not targeted to specific subregions of IT cortex known to play a causal role in discrimination of the visual object class being studied[21] and where the visual feature code is precisely understood[3]. Here, to study the neural mechanism that represents long-term object memories, we targeted three regions: anterior medial face patch (AM), the most anterior face patch in IT cortex[22], and PR and TP, two recently reported face patches in the perirhinal cortex and anterior temporal pole, respectively[23,24]. These three regions lie at the apex of the macaque face patch system, an anatomically connected network of regions in the temporal lobe dedicated to face processing[22,25–29]. AM harbours a strong signal for invariant facial identity[3,22], perirhinal cortex is known to play a critical role in visual memory[30–33] and TP has recently been suggested to provide a privileged pathway for rapid recognition of familiar individuals[24]. We thus hypothesized that a representation of face memory should occur in AM, PR and/or TP.

Our recordings showed that, in all three patches, familiar faces were distinguished from unfamiliar faces. First, in all three patches, familiar faces were represented in a subspace distinct from unfamiliar faces. Second, in all three patches the relative response magnitude to familiar faces differed significantly from that to unfamiliar faces; however, the sign of this difference was not stable and depended strongly on the relative frequency of presentation of familiar and unfamiliar faces (that is, temporal context). Third, and most strikingly,

[1]Division of Biology and Biological Engineering, Caltech, Pasadena, CA, USA. [2]Mortimer B. Zuckerman Mind Brain Behavior Institute, Columbia University, New York City, NY, USA. [3]Neurobiology Section, Division of Biological Sciences, University of California, San Diego, San Diego, CA, USA. [4]Howard Hughes Medical Institute, University of California, Berkeley, CA, USA. [5]Present address: Department of Neuroscience, University of California, Berkeley, CA, USA. ✉e-mail: liangshe@caltech.edu; dortsao@berkeley.edu

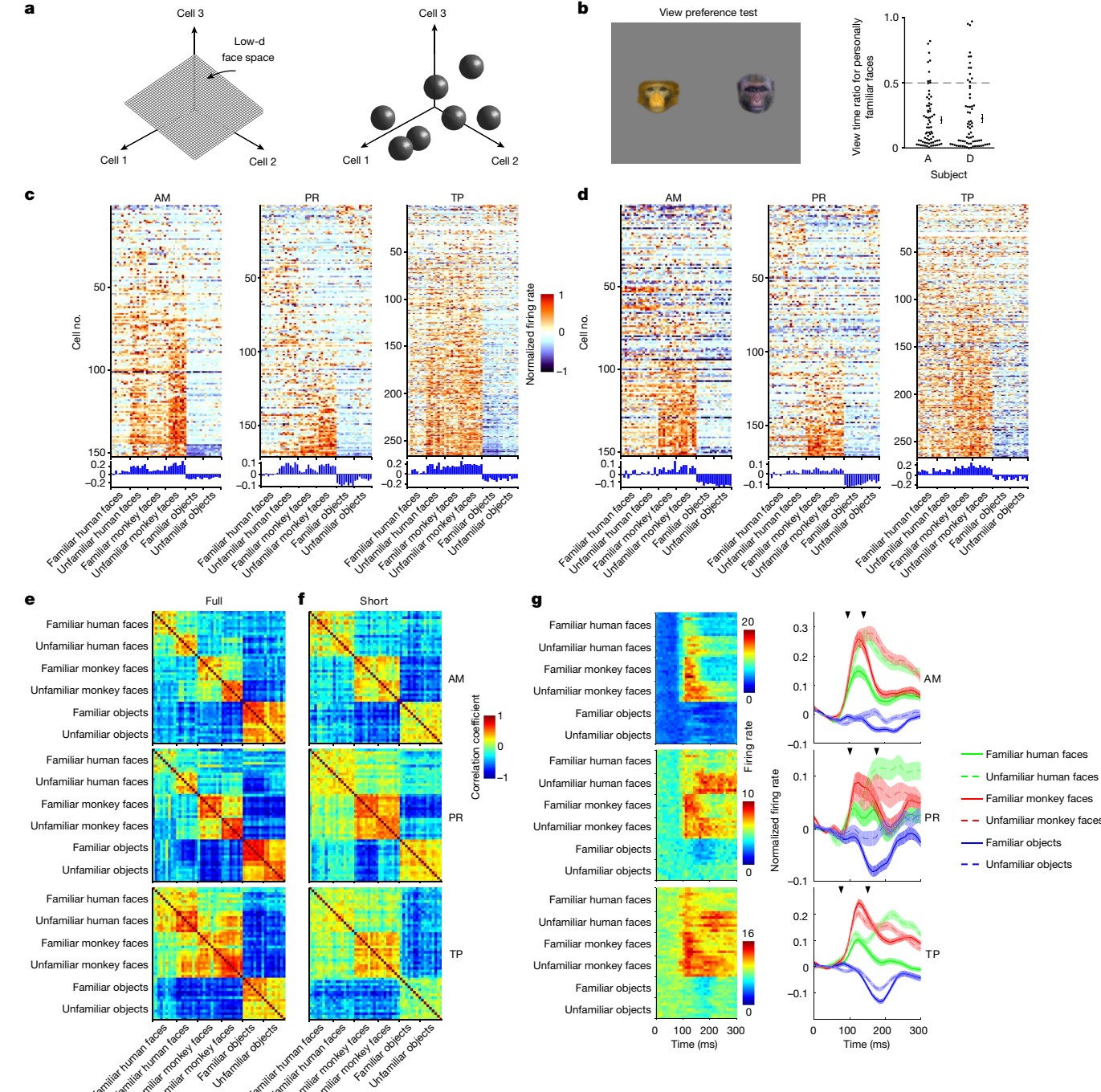

**Fig. 1 | Cells in face patches are modulated by familiarity. a**, Two alternative schemes for face representation: a low-dimensional (low-d) continuous feature space (left) and a set of discrete attractors (right). **b**, Left, view preference test. Pairs of faces (*n* = 72), one familiar and one unfamiliar, were presented for 10 s and the time spent fixating each was recorded. Right, ratio of time spent fixating personally familiar versus unfamiliar faces for two animals (each dot represents one face pair). Error bar, mean ± s.e.m. **c**, Responses of cells to stimuli from six stimulus categories (familiar human faces, unfamiliar human faces, familiar monkey faces, unfamiliar monkey faces, familiar objects and unfamiliar objects) across three face patches (AM, PR and TP). Responses were averaged between 50 and 300 ms following stimulus onset ('full'-response window; AM, *n* = 152 cells; PR, *n* = 171 cells; TP, *n* = 266 cells; Supplementary Methods and Supplementary Table 1 provide additional statistical information). **d**, Same as **c** but for the 'short'-response window

(AM, 50–125 ms; PR, 50–150 ms; TP, 50–150 ms). **e**, Similarity (Pearson correlation coefficient) matrix of population responses for full-response window. **f**, Same as **e** but for short-response window. **g**, Left, average response time course across AM (top), PR (middle) and TP (bottom) populations to each of the screening stimuli. Right, AM, PR and TP response time courses averaged across both cells and category exemplars (normalized for each cell; Supplementary Methods). Left-hand arrow indicates mean time when visual responses to faces became significantly higher than baseline (AM, 85 ms; PR, 105 ms; TP, 75 ms; Supplementary Methods); right-hand arrow indicates mean time when responses to familiar versus unfamiliar faces became significantly different (AM, 105 and 145 ms for human and monkey faces, respectively; PR, 155 and 215 ms, respectively; TP, 145 and 205 ms, respectively). Shaded areas, s.e.m. across neurons.

in AM and PR, but not in TP, familiar faces were encoded by a unique geometry at long latency; furthermore, unlike response magnitude, this unique geometry associated with familiar faces was stable across contexts. These results suggest that the memory of familiar faces is primarily represented in face patches AM and PR through axis change rather than altered response magnitude. This conclusion—that a major piece of the network code for visual memory is temporally multiplexed with the perceptual code and activated only at long latency—sheds light on how we can both veridically perceive visual stimuli and recall past experiences from them using the same set of neurons.

## AM and PR are modulated by familiarity

We identified face patches AM, PR and TP in five animals using functional magnetic resonance imaging[25]. To characterize the role of familiarity in modulating the responses of cells in AM, PR and TP, we targeted electrodes to these three patches (Extended Data Fig. 1) and recorded responses to a set of screening stimuli consisting of human faces, monkey faces and objects. The stimuli were either personally familiar or unfamiliar (Extended Data Fig. 2a), with eight or nine images per category. Personally familiar images depicted people, monkeys and objects with which the animals interacted on a daily basis; a new set of unfamiliar images was presented per recording site. Animals showed highly significant preferential looking towards the unfamiliar face stimuli and away from familiar face stimuli (Fig. 1b), confirming behaviourally that these stimuli were indeed familiar to the monkey[34]. Monkeys also performed significantly better on a face identification task for familiar compared with unfamiliar faces (Extended Data Fig. 3a,b), indicating a behavioural recognition advantage for familiar faces.

Across the population, 93% of cells in AM, 74% in PR and 88% in TP were face selective (Extended Data Fig. 3c). Below, we group data from three monkeys for AM, three for PR and two for TP becaue we did not find any marked differences between individuals (Extended Data Figs. 4 and 5 show the main results separately for each animal). All three patches exhibited a significantly stronger response across the population to unfamiliar compared with personally familiar stimuli in this experiment (Fig. 1c). This is inconsistent with a recent study reporting that TP is specialized for representing personally familiar faces[24] (however, the latter study never actually presented unfamiliar faces but contrasted responses only to personally versus pictorially familiar faces; Extended Data Fig. 6a–c provides further detail). Further casting doubt on a specialized role for TP in encoding personally familiar faces, we found that the response in TP to faces of other species was stronger than to human or monkey faces (Extended Data Fig. 6d–f). Overall, the pattern of decreased responses to familiar faces across AM, PR and TP is consistent with a large number of previous studies reporting suppression of responses to familiar stimuli in IT and perirhinal cortex[5–10]. Individual cells showed a diversity of selectivity profiles for face species and familiarity type (Extended Data Fig. 7a–c). Representation similarity matrices showed distinct population representations of the six stimulus classes in both AM and PR, and more weakly in TP (Fig. 1e).

Mean responses to familiar versus unfamiliar faces diverged over time, with difference becoming significant at 125 ms in AM, 185 ms in PR and 175 ms in TP; the mean visual response to faces themselves significantly exceeded baseline earlier, at 85 ms in AM, 105 ms in PR and 75 ms in TP (Fig. 1g). The delay in suppression to familiar faces is consistent with previous reports of delayed suppression to familiar stimuli in IT[5,7–9]. Single-cell response profiles and representation similarity matrices computed using a short time window showed less distinct responses to familiar versus unfamiliar stimuli (Fig. 1d,f). Overall, the results so far show that AM, PR and TP all exhibit long-latency suppression to familiar faces.

## An axis code for unfamiliar faces

Responses of AM, PR and TP cells to familiar stimuli, although lower on average at long latencies, remained highly heterogeneous across faces (Fig. 1c and Extended Data Fig. 7a–c), indicating that cells were driven by both familiarity and identity. We next asked how familiarity interacts with the recently discovered axis code for facial identity[3].

According to this axis code, face cells in IT compute a linear projection of incoming faces formatted in shape and appearance coordinates onto specific preferred axes[3]. For each cell, the preferred axis is given by the coefficients $\mathbf{c}$ in the equation $r = \mathbf{c} \cdot \mathbf{f} + c_0$, where $r$ is the response of the cell, $\mathbf{f}$ is a vector of shape and appearance features and $c_0$ is a constant offset (Supplementary Methods); shape features capture variations in the location of key facial landmarks (for example, outline, eye, nose and mouth positions and so on) whereas appearance features capture the shape-independent texture map of a face[3]. Together, a population of face cells with different preferred axes encodes a face space that is embedded as a linear subspace of the neural state space. The axis code has so far been examined only for unfamiliar faces. By studying whether and how this code is modified by familiarity, we reasoned that we could potentially understand the code for face memory.

We first asked whether face cells encode familiar and unfamiliar faces using the same axis. To address this, we examined tuning to unfamiliar faces (described in this section) and then compared this with tuning to familiar faces (described in the next section). We began by mapping the preferred axes of AM, PR and TP cells using a set of 1,000 unfamiliar monkey faces (Extended Data Fig. 2b). We used monkey faces because responses to the screening stimuli were stronger to monkey than to human faces on average in AM/PR/TP (Fig. 1c; $P < 4 \times 10^{-6}$, two-sided paired $t$-test, $t = -4.68$, degrees of freedom = 588, difference = 0.75 Hz, 95% confidence interval = [0.44, 1.07], $n = 589$ cells pooled across AM, PR and TP). The 1,000 monkey faces were randomly drawn from a monkey face space defined by 120 parameters (Supplementary Methods) encompassing a wide variety of identities, allowing the selection of a subset that was matched in feature distributions to familiar faces (Extended Data Fig. 8).

As expected, cells in AM showed ramp-shaped tuning along their preferred axes (Fig. 2a and Extended Data Fig. 3e). Interestingly, a large proportion of cells in PR and TP also showed ramp-shaped tuning along their preferred axes (Fig. 2a and Extended Data Fig. 3e). To our knowledge this is the first time that axis coding of visual features has been reported for face patches outside the IT cortex. In all three patches, preferred axes computed using split halves of the data were highly consistent (Extended Data Fig. 3f). These results suggest that AM, PR and TP share a common axis code for representing unfamiliar faces.

## Off-axis responses to familiar faces

We next examined how familiarity modulates the axis code. We projected the features of personally familiar and a random subset of unfamiliar faces onto the preferred axis of each AM/PR/TP cell and plotted responses. In AM and PR, responses to unfamiliar faces followed the axis (Fig. 2a, green dots) whereas, strikingly, responses to familiar faces departed from the axis (Fig. 2a, yellow dots).

This departure in AM and PR was not a simple gain change: the strongest responses to familiar faces were often to faces projecting somewhere in the middle of the ramp rather than on the end (Fig. 2a). It cannot be explained, therefore, by an attentional increase or decrease to familiar faces, which would elicit a gain change[35]. Indeed, the effect cannot be explained by any monotonic transform in response, such as repetition suppression or monotonic sparsening[8,10], because any such transform should preserve the rank ordering of preferred stimuli.

The surprising finding of off-axis responses to familiar faces was prevalent across the AM and PR populations, but not TP. To quantify

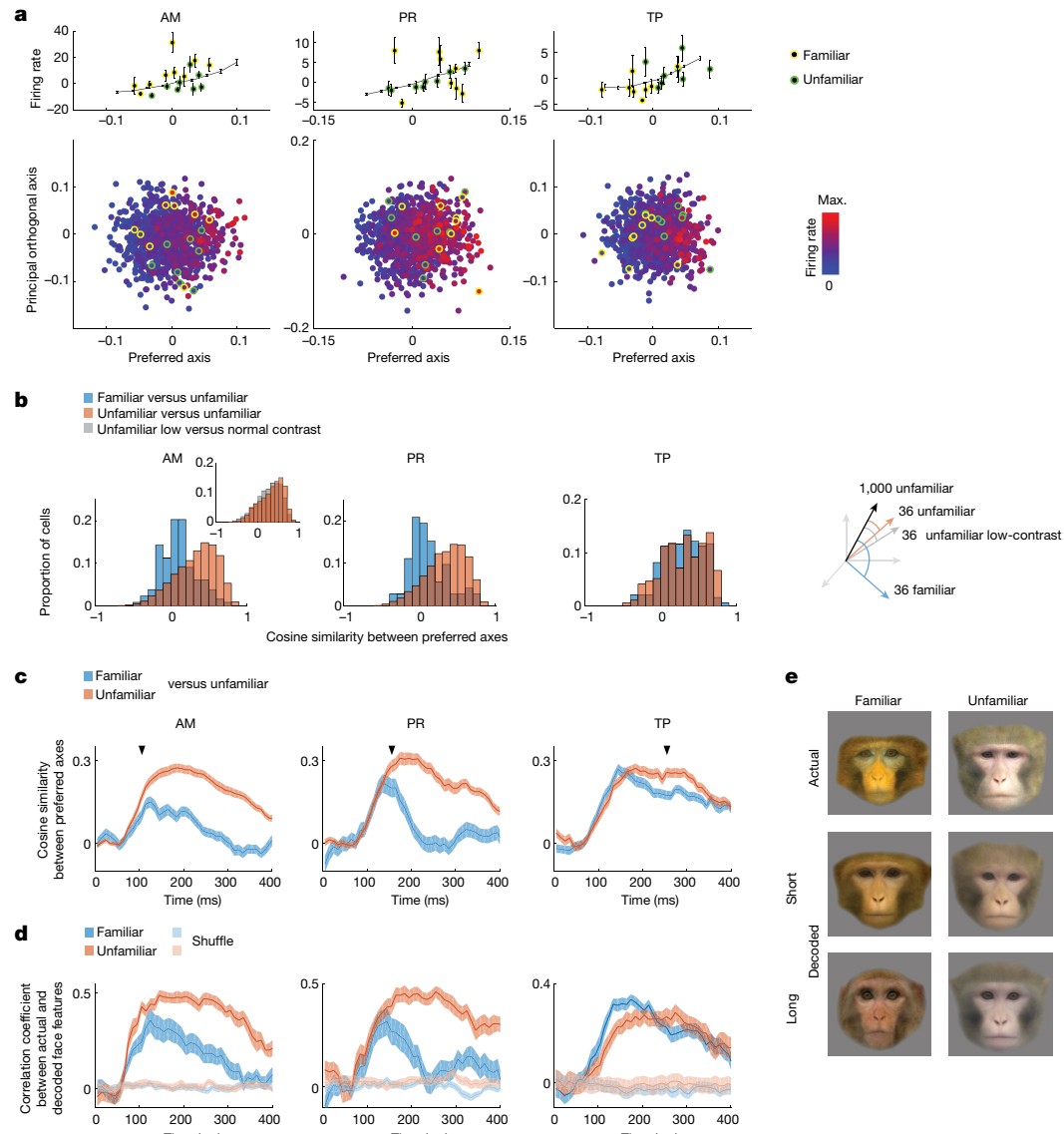

**Fig. 2 | AM and PR cells use different axes to represent familiar versus unfamiliar faces. a**, Three example cells showing axis tuning. Top, mean response as a function of distance along the preferred axis. Green (yellow) dots denote responses to eight random unfamiliar (nine personally familiar) faces. Error bar, s.e.m. Bottom, responses to 1,000 unfamiliar faces, projected onto the cell's preferred axis and principal (longest) orthogonal axis in the face feature space. Response magnitudes are colour coded. **b**, Population analysis comparing preferred axes for familiar versus unfamiliar faces. Distribution of cosine similarities between axes computed using 1,000 − 36 unfamiliar faces and 36 omitted unfamiliar faces (orange), and between axes computed using 1,000 − 36 unfamiliar faces and 36 familiar faces (blue) are shown. Preferred axes were computed using the top ten shape and top ten appearance features of presented faces. Inset, control experiment with 36 low-contrast faces in

place of 36 familiar faces. **c**, Time course of cosine similarity between preferred axes for unfamiliar–unfamiliar (orange) and unfamiliar–familiar (blue) faces as in **b**. Arrowheads indicate when differences became significant (AM, 105 ms; PR, 175 ms; TP, 255 ms; one-tailed *t*-test, *P* < 0.001; AM, *n* = 134 cells; PR, *n* = 72 cells; TP, *n* = 197 cells; Supplementary Methods and Supplementary Table 1 give additional information about statistical tests). Shaded areas, s.e.m. **d**, Time course of linear decoding of facial features (Supplementary Methods). Shaded areas, s.e.m. Lighter colour, same analysis using stimulus identity-shuffled data (ten repeats). **e**, Example linearly reconstructed faces from short (120–170 ms) and long (220–270 ms) latency responses combining cells from both AM and PR. Reconstructions were performed using linear decoders trained on a large set of unfamiliar faces, similar to **d** (Supplementary Methods).

this phenomenon at the population level we first created a larger set of familiar faces. To this end, animals were shown face images and videos daily for at least 1 month, resulting in a total of 36 familiar monkey faces, augmenting the nine personally familiar monkey faces in our initial screening set (Extended Data Fig. 2c and Supplementary Methods). Preferential looking tests confirmed that pictorially and cinematically familiar faces were treated similarly to the personally familiar faces (Extended Data Fig. 3d). These 36 familiar faces were presented randomly interleaved with the 1,000 unfamiliar monkey faces while we recorded from AM, PR and TP.

We computed preferred axes for cells using responses to the 36 familiar faces. We found that, when familiar and unfamiliar faces were matched in number (36), familiar axes performed as well in explaining responses to familiar faces as unfamiliar axes in explaining responses to unfamiliar faces (Extended Data Fig. 3g,h). The comparable strength of axis tuning for familiar and unfamiliar faces naturally raised the question: are familiar and unfamiliar axes the same?

To compare familiar and unfamiliar axes, for each cell we first computed the preferred axis using responses to the large set of unfamiliar faces (1,000 − 36 faces). We then correlated this to a preferred axis

computed using responses to either (1) the set of 36 familiar faces ('unfamiliar–familiar' condition) or (2) the omitted set of 36 unfamiliar faces ('unfamiliar–unfamiliar' condition). The distribution of correlation coefficients showed significantly higher similarities for the unfamiliar–unfamiliar compared with the unfamiliar–familiar condition in AM and PR, but not in TP (Fig. 2b).

As a control, we presented a set of low-contrast faces expected to elicit a simple decrease in response gain but preserving rank ordering of preferred stimuli. Confirming expectations, axis similarities computed using these contrast-varied faces were not significantly different for high–high- versus high–low-contrast faces (Fig. 2b, inset). As a second control, to ensure that the effects were not due to differences in the feature content of familiar versus unfamiliar faces, we identified 30 familiar and 30 unfamiliar faces that were precisely feature matched. In brief, we used gradient descent to search for a subset of familiar and unfamiliar faces that were matched in the distribution of each feature as well as in the distribution of pairwise face distances (Supplementary Methods and Extended Data Fig. 8). We recomputed unfamiliar–familiar and unfamiliar–unfamiliar correlations and continued to find that familiar faces were encoded by a different axis than unfamiliar faces in AM and PR, but not in TP (Extended Data Fig. 9a, top). Finally, we confirmed that axis divergence persisted when axes were computed using only the subset of cells showing significant axis tuning for both familiar and unfamiliar faces (Extended Data Fig. 9a, middle).

Previously we observed that the decrease in firing rate for familiar faces occurred at long latency (Fig. 1g). We next investigated the time course of the deviation in the preferred axis. We performed a time-resolved version of the analysis in Fig. 2b, comparing the preferred axis computed from 36 unfamiliar or 36 familiar faces with that computed from $1,000 - 36$ unfamiliar faces over a rolling time window (Fig. 2c). Initially, axes for familiar and unfamiliar faces were similar but, at longer latency ($t > 105$ ms in AM, $t > 155$ ms in PR), the preferred axis for familiar faces diverged from that for unfamiliar faces.

The divergence in preferred axis over time for familiar versus unfamiliar faces suggests that the brain would need to use a different decoder for familiar versus unfamiliar faces at long latencies. Supporting this, in both AM and PR, at short latencies, feature values for familiar faces obtained using a decoder trained on unfamiliar faces matched actual feature values, and reconstructions were good (Fig. 2d,e). By contrast, a decoder trained on unfamiliar faces at long latency performed poorly on recovering feature values of familiar faces (Fig. 2d,e).

Could the apparent axis change be explained by a simpler change— for example, sensitivity decrease in a subset of features or an output nonlinearity change, without necessitating a change in axis? Further analyses demonstrated that these simpler models could not explain the change in responses of cells to familiar faces (Extended Data Fig. 9b,c).

## An early shift in familiar face subspace

So far we have uncovered a distinct geometry for encoding familiar versus unfamiliar face features in AM and PR at long latency. But how is the categorical variable of familiarity itself encoded in AM and PR? Previous studies have suggested that familiarity is encoded by response suppression across cells[5–10]. Supporting this, our first experiment (a screening set consisting of familiar and unfamiliar human faces, monkey faces and objects) showed a decreased average response to familiar compared with unfamiliar faces (Fig. 1). However, to our great surprise, data from our second experiment (1,000 unfamiliar faces interleaved with 36 familiar faces; Fig. 2) showed a stronger mean response to familiar compared with unfamiliar stimuli (Fig. 3a,b). This was true even when we compared responses to the exact same subset of images (Extended Data Fig. 9d). What could explain this reversal? The two experiments had one major difference: in the first experiment the ratio of familiar to unfamiliar faces was 34:16 whereas in the second the ratio was 36:1,000 (in both experiments, stimuli were randomly interleaved and presentation times were identical). Thus the expectation of familiar faces was much lower in the second experiment. Previous studies in IT have suggested that expectation can strongly modulate response magnitudes, with unexpected stimuli exhibiting stronger responses[36]. The marked reversal of relative response magnitude to familiar versus unfamiliar faces across the two experiments suggests that mean response magnitude is not a robust indicator of familiarity, because it depends on temporal context. Importantly, and, by contrast, axis change for familiar faces was stable across the two experiments (Extended Data Fig. 9e).

Even more challenging to the repetition suppression model of familiarity coding, the accuracy for decoding familiarity rose above chance extremely early, starting at 95 ms in AM, 105 ms in PR and 135 ms in TP (Fig. 3c, decoding using responses from Experiment 2); in PR this occurred even before any significant difference in mean firing rates between familiar and unfamiliar faces (compare black arrowheads in Fig. 3c with green arrowheads in Fig. 3b). What signal could support this ultrafast decoding of familiarity, which moreover generalized across face identity, if not mean firing rate difference? Recall earlier that we had found that, at short latency, familiar faces were encoded using the same axes as unfamiliar faces (Fig. 2d). This implies that, at short latency, familiar and unfamiliar faces are represented in either identical or parallel manifolds. Agreeing with this, familiar face features could be readily decoded using a decoder trained on unfamiliar faces (Fig. 2e). This suggested to us that their representations might be shifted relative to each other and that this shift is what permits early familiarity decoding. A plot of the neural distance between familiar and unfamiliar response centroids over time supported this hypothesis (Fig. 3d): the familiar–unfamiliar centroid distance increased extremely rapidly compared with that of unfamiliar–unfamiliar, and $d'$ (Supplementary Methods) along the unfamiliar–familiar centroid axis became significantly higher than a shuffle control at 95 ms in AM, 105 ms in PR and 135 ms in TP, equal to the time when familiarity could be decoded significantly above chance in each of these areas. Direct inspection of shifts between responses to familiar versus unfamiliar faces across cells showed a distribution of positive and negative values that could be exploited by a decoder for familiarity (Fig. 3e).

Further supporting the shift hypothesis, we found that the familiarity decoding axis was orthogonal to the face feature space at both short and long latency. We computed cosine similarity in the neural state space between the familiarity decoding and face feature decoding axes, both familiar and unfamiliar, for 20 features capturing the most variance. The resulting values were tightly distributed around 0 at both short (50–150 ms) and long (150–300 ms) latency (Fig. 3f). Overall, these results suggest a geometric picture in which familiar and unfamiliar stimuli are represented in distinct subspaces, with the familiar face subspace shifted relative to the unfamiliar face subspace at short latencies and then further distorted at long latencies in AM and PR (Fig. 3g).

## Localization of the site of face memory

The finding of memory-driven axis change at long latency in AM and PR is consistent with decades of functional studies suggesting a unique role for interactions between IT and the medial temporal lobe in memory formation[37,38]. Is the distinct representational geometry for familiar faces at long latency in AM due to feedback from PR? To address this we silenced PR while recording responses to familiar and unfamiliar faces in AM (Fig. 4a). IT cortex is known to receive strong feedback from perirhinal cortex[39], and this is true in particular for face patch AM[29]. Consistent with this, inactivation of PR produced strong changes in AM responses with some cells showing an increase in response and others a decrease (Fig. 4b,c).

We next asked whether feedback modulation from PR specifically affected AM responses to familiar faces, as one might expect if PR were the source of AM memory signals. We found that divergence between familiar and unfamiliar axes at long latency continued to occur in AM

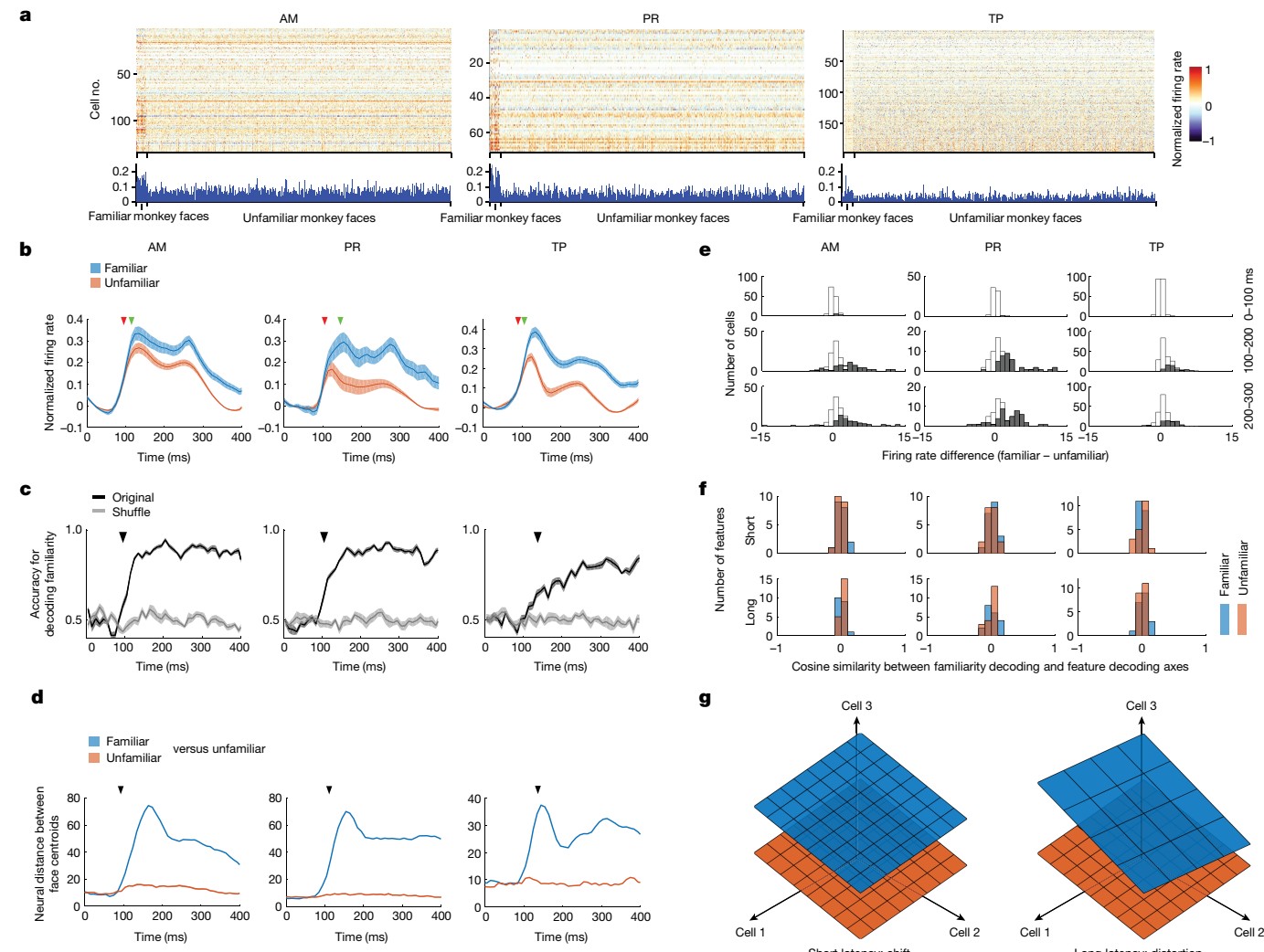

**Fig. 3 | An early shift in response subspace allows decoding of familiarity.**
**a**, Responses of cells to stimuli from 36 familiar and 1,000 unfamiliar monkey
faces, averaged over 50–300 ms following stimulus onset. **b**, Response time
course across AM, PR and TP populations, averaged across cells and all
familiar or unfamiliar faces from the 1,036 monkey face stimulus set. Shaded
areas, s.e.m. Red arrowheads indicate the time when responses to faces
became significantly higher than baseline (AM, 95 ms; PR, 105 ms; TP. 90 ms;
Supplementary Methods). Green arrowheads indicate the time when responses
to familiar versus unfamiliar faces became significantly different (AM, 125 ms;
PR, 135 ms; TP, 105 ms). **c**, Black line, time course of accuracy for decoding
familiarity; shaded area, s.e.m.; grey line, chance level. Arrowheads indicate the
time at which decoding accuracy rose above chance (AM, 95 ms; PR, 105 ms;
TP, 135 ms). **d**, Time course of neural distance between centroids of 36 familiar

and 1,000 − 36 unfamiliar face responses (blue) and between centroids of
responses to a subset of 36 unfamiliar faces and responses to the remaining
1,000 − 36 unfamiliar faces (orange). Arrowheads indicate the time when $d'$
along the two centroids became significant (AM, 95 ms; PR, 105 ms; TP, 135 ms).
**e**, Distribution of differences between mean firing rates to familiar and
unfamiliar faces at three different time intervals. Grey bars indicate cells
showing a significant difference (Supplementary Methods). **f**, Distribution of
cosine similarities between familiarity decoding and face feature decoding axes
at short (50–150 ms) and long (150–300 ms) latency for the first 20 features
(ten shape, ten appearance). **g**, Schematic illustration of neural representation
of familiar (blue) and unfamiliar (orange) faces at short and long latency for
AM and PR.

following PR inactivation (Fig. 4d). Indeed, responses to familiar and
unfamiliar faces were similarly modulated by PR inactivation across the
population (Fig. 4e). Finally, decoding of both face familiarity and face
features from AM activity was unaffected by PR inactivation (Fig. 4f,g).
Overall, these results show that inactivation of PR had a strong effect
on the gain of AM responses but no apparent effect on face coding,
including memory-related axis change.

Do signatures of familiarity coding, as observed in AM, PR and TP,
exist even earlier in the face patch pathway? We mapped responses to
familiar and unfamiliar faces in middle lateral face patch (ML), a hier-
archically earlier patch in the macaque face-processing pathway that
provides direct anatomical input to AM[22,29]. Responses to the screening
stimuli in ML exhibited a similar pattern as in AM, showing suppression

to personally familiar faces at long latency (Extended Data Fig. 10a,c).
However, population representation similarity matrices did not show
distinct responses to familiar versus unfamiliar faces (Extended Data
Fig. 10b). Furthermore, the population average firing rate showed a sus-
tained divergence between responses to familiar and unfamiliar faces
much later than in AM (160 compared with 140 ms in AM; Extended Data
Fig. 10c), suggesting that ML may receive a familiarity-specific feedback
signal from AM. Importantly, ML neurons also showed axis divergence
(Extended Data Fig. 10d–f), consistent with the idea that memory is
stored in a distributed way across the entire hierarchical network used
for representation[40]. Finally, familiarity could be decoded in ML even
earlier than in AM (Extended Data Fig. 10g–i). Overall, these results sug-
gest that ML also plays a significant role in storing memories of faces.

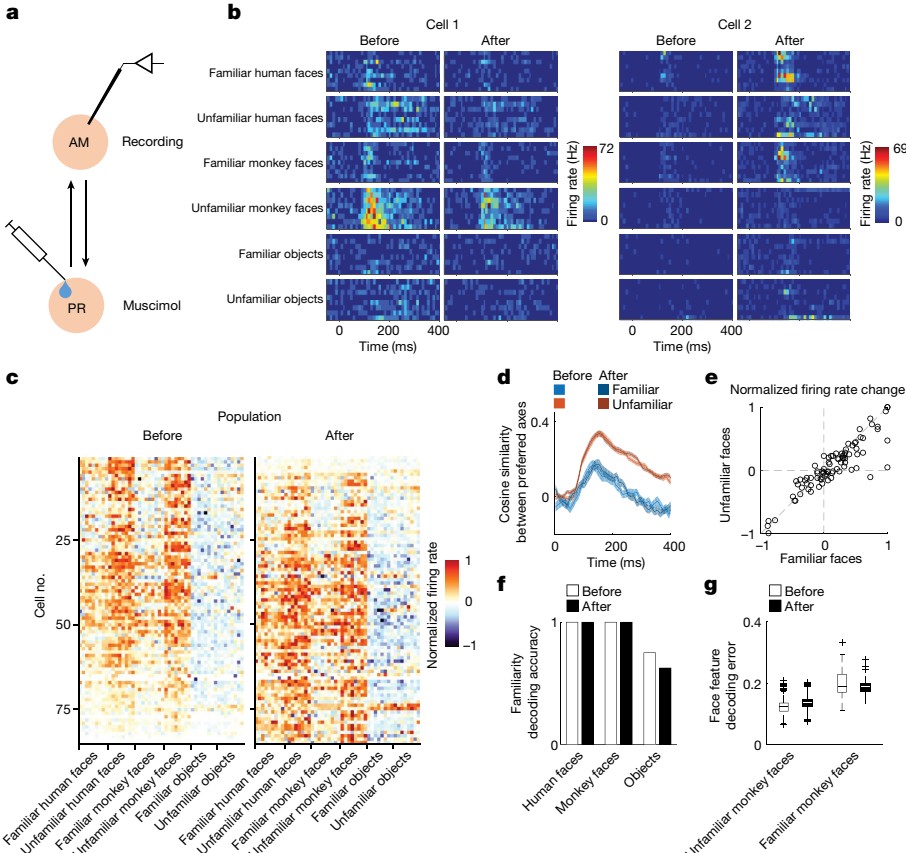

**Fig. 4 | Axis change for familiar faces does not depend on PR feedback to IT.**
**a**, Schematic of experiment aimed at identifying the origin of memory-related signals in AM. Muscimol was injected into face patch PR and the responses of AM neurons were recorded both before and after PR inactivation using a multi-electrode probe. **b**, Responses of two example AM cells to screening stimuli before and after PR inactivation. **c**, Response profiles of the AM population to screening stimuli before and after PR inactivation. **d**, Time course of the similarity between preferred axes for unfamiliar–unfamiliar and unfamiliar–familiar faces (as in Fig. 2c) before and after PR inactivation. Shaded areas, s.e.m. **e**, Normalized firing rate changes for familiar and unfamiliar faces induced by PR inactivation ($n = 85$ cells). **f**, Familiarity decoding accuracy for screening stimuli before and after PR inactivation ($n = 17$ for human and monkey faces, $n = 16$ for objects). For objects, chi-square test $\chi^2(1, N = 16) = 0.58$, $P = 0.45$. **g**, Face feature decoding error (mean square error) for unfamiliar ($n = 1,000$) and familiar ($n = 36$) faces before and after PR inactivation.

## Discussion

In this paper we investigated the elusive neural code for long-term object memory. Although classic lesion studies suggest that long-term object memories should reside in IT cortex[1], recent work on IT coding has focused on representation of incoming visual input and concluded that IT neurons extract high-level visual features agnostic to semantic content[4,41]. How can such meaning-agnostic, feature-selective cells be responsible for encoding long-term object memories that are highly context and familiarity dependent? Here we shed light on this conundrum, finding that in anterior face patches AM and PR a distinct neural code for familiar faces emerges at long latency in the form of a change in preferred axis. Thus, feedforward feature-coding properties of IT cells may be reconciled with a putative role in long-term memory through temporal multiplexing. Inactivation of PR did not affect axis change dynamics in AM, suggesting that the memory-related axis change mechanism may be intrinsic to IT cortex.

Previous physiological work on representation of familiar stimuli has focused largely on repetition suppression, the observation that the response to familiar stimuli is reduced[5–11]. We found that repetition suppression was not a robust indicator of familiarity in any face patch. Instead, relative response amplitude to familiar versus unfamiliar faces was highly sensitive to temporal context. We speculate that these relative response amplitudes, and associated neural distances and decoding accuracies (Extended Data Fig. 10j,k), may reflect momentary changes in stimulus saliency rather than face memory. By contrast, axis change for familiar faces at long latency was consistent across context (Extended Data Fig. 9e), indicating a reliable code for face memory.

What could the computational purpose of this axis change be? We speculate that, by lifting representations of face memories into a separate subspace from that used to represent unfamiliar faces (Fig. 3g), attractor-like dynamics may be built around these memories through a recurrent network to allow reconstruction of familiar face features from noisy cues without interfering with veridical representation of sensory inputs[42,43]. Computational considerations make it clear that the ability to recall (that is, reconstruct from noisy cues) a large number of familiar faces requires a code change. This is because a perfectly disentangled representation (the axis code) is inherently low dimensional; the memory capacity of a recurrent network using disentangled representations increases only linearly with the number of dimensions of the representation[44]. Importantly, recoding stimuli with small, nonlinear distortions of disentangled representations can significantly increase the memory capacity to one that scales linearly with the number of neurons[43,44], as in Hopfield networks with random memories[43]. We hypothesize that long-latency axis change reflects this recoding. To date, studies of IT have emphasized the stability of response tuning over months[45,46]. Our results suggest such stability coexists with a precisely orchestrated dynamics for representing familiar stimuli through the mechanism of long-latency change in axis.

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

## Reporting summary

Further information on research design is available in the Nature Portfolio Reporting Summary linked to this article.

## Data availability

The dataset of neural responses to screening stimuli and 1,000 monkey faces is available at https://doi.org/10.5281/zenodo.10460607 (ref. 47). Other datasets are available from the PrimFace database (http://visiome.neuroinf.jp/primface), FERET database (https://www.nist.gov/itl/products-and-services/color-feret-database), CVL Face Database (http://www.lrv.fri.uni-lj.si/facedb.html), MR2 face database (https://osf.io/skbq2/), Chicago Face Database (https://www.chicagofaces.org/), CelebA CelebFaces Attributes Dataset (https://mmlab.ie.cuhk.edu.hk/projects/CelebA.html), FEI Face Database (https://fei.edu.br/~cet/facedatabase.html), PICS Psychological Image Collection at Stirling (https://pics.stir.ac.uk), Caltech faces 1999 (https://data.caltech.edu/records/6rjah-hdv18), Essex Face Recognition Data (http://cswww.essex.ac.uk/mv/allfaces/faces95.html), and The MUCT Face Database (www.milbo.org/muct). Source data are provided with this paper.

## Code availability

The code that reproduces the core results (Fig. 2b,c and Extended Data Fig. 5a,b) is available at https://doi.org/10.5281/zenodo.10460607 (ref. 47). All other code is available from the corresponding authors on reasonable request.

47. She, L., Benna, M., Shi, Y., Fusi, S., & Tsao, D. Data and code for "Temporal multiplexing of perception and memory codes in IT cortex. She et al. Nature 2024". *Zenodo* https://doi.org/10.5281/zenodo.10460607 (2024).

**Acknowledgements** This work was supported by NIH (nos. DP1-NS083063 and EY030650-01), the Howard Hughes Medical Institute, the Simons Foundation, the Human Frontiers in Science Program, the Office of Naval Research and the Chen Center for Systems Neuroscience at Caltech. S.F. is supported by the Simons Foundation, the Gatsby Charitable Foundation, the Swartz Foundation and the NSF's NeuroNex Program (award no. DBI-1707398). We thank K. M. Gothard for sharing monkey face images, D. Chung and V. Tong for assistance with behavioural testing and N. Schweers for assistance with animal training and scanning.

**Author contributions** L.S. and D.Y.T. conceived the project and designed experiments. L.S. and Y.S. collected data. L.S. and M.K.B. analysed data. L.S., M.K.B. and D.Y.T. interpreted data, with feedback from S.F. L.S. and D.Y.T. wrote the paper, with feedback from S.F., M.K.B. and Y.S.

**Competing interests** The authors declare no competing interests.

**Additional information**
**Correspondence and requests for materials** should be addressed to Liang She or Doris Y. Tsao.

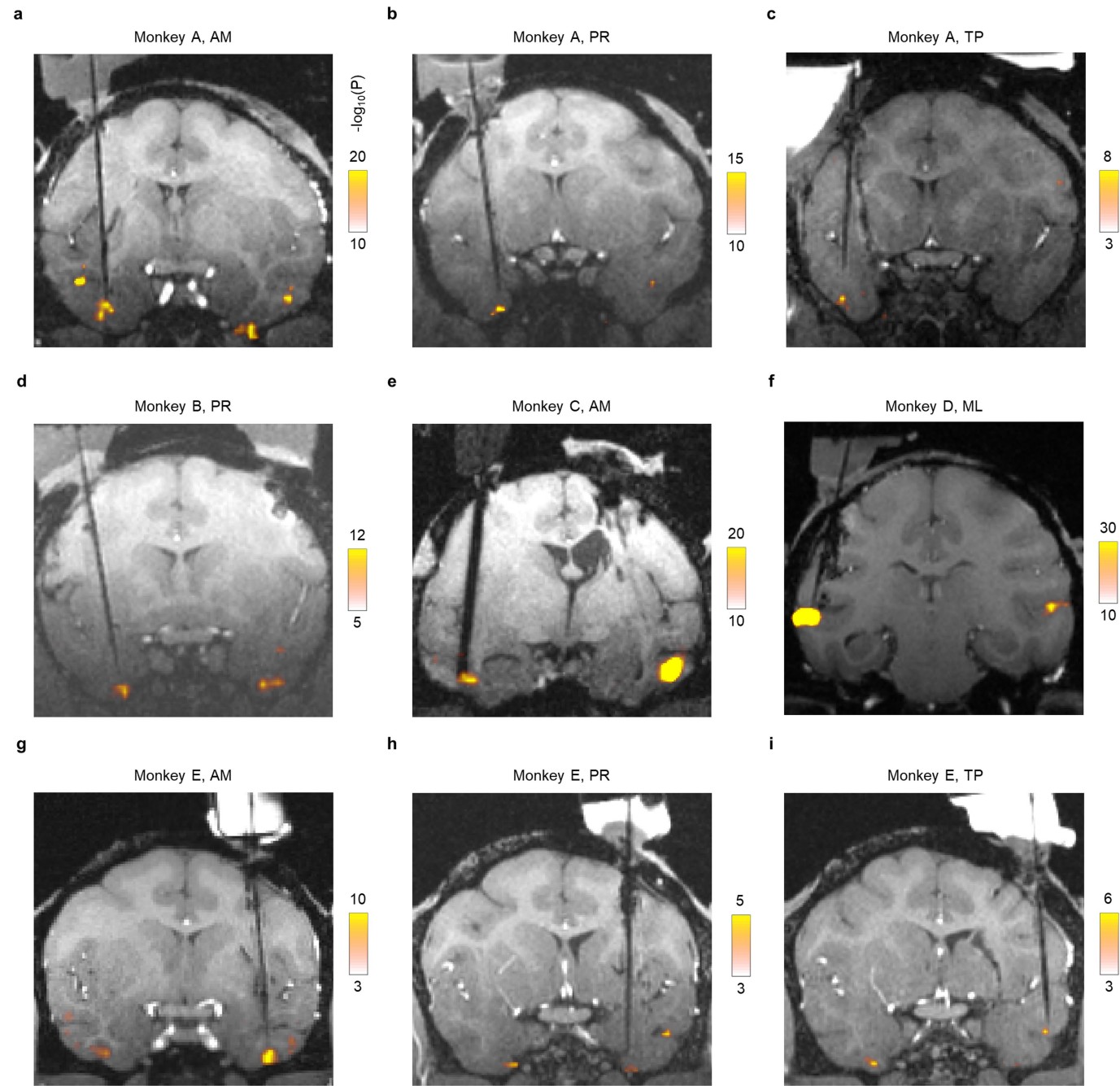

**Extended Data Fig. 1 | Coronal slices showing the electrode targeting nine recording sites from five monkeys. a**–**c**, Single electrode targeting AM, PR, and TP in monkey A. **d**, Single electrode targeting PR in monkey B. **e**, Brush array electrodes targeting AM in monkey C. **f**, Single electrode targeting ML in monkey D. **g**–**i**, Single electrode targeting AM, PR, and TP in monkey E. Activations for the contrast faces versus objects are shown, at *p* values in −log10, two-sided t-test, not corrected for multiple comparisons. Note: There is no corresponding MRI image showing the recording targeting ML in monkey A because the recording was performed using an early version of an Neuropixels NHP probe which was not MRI compatible.

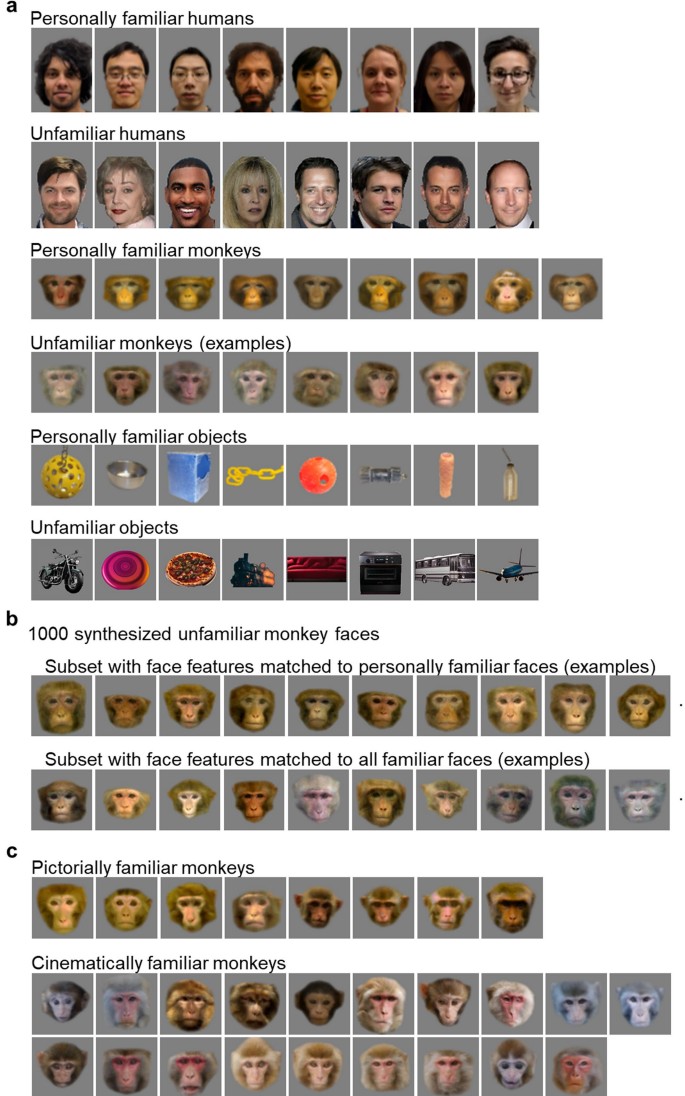

**a**

Personally familiar humans

Unfamiliar humans

Personally familiar monkeys

Unfamiliar monkeys (examples)

Personally familiar objects

Unfamiliar objects

**b**

1000 synthesized unfamiliar monkey faces

Subset with face features matched to personally familiar faces (examples)

Subset with face features matched to all familiar faces (examples)

**c**

Pictorially familiar monkeys

Cinematically familiar monkeys

**Extended Data Fig. 2 | Visual stimuli. a**, Screening stimuli. Eight out of nine personally familiar faces are shown. Example unfamiliar stimuli are shown here; a new set was presented for every recording site, drawn from image sets described in the Methods. Note that unfamiliar human faces and unfamiliar objects are not the actual stimuli but synthetic images similar to the actual stimuli, due to difficulty in obtaining permission for publication. **b**, Examples of unfamiliar faces in the thousand face stimulus set. Monkey faces were generated by a 120d shape-appearance model (see Methods). The thousand monkey face stimulus set was extremely diverse, allowing subsets of faces to be chosen that were matched in feature distributions to familiar faces (see Supplementary Methods). Shown here are examples from two subsets, one matched to the personally familiar faces, and one matched to all familiar faces. **c**, Additional familiar faces (pictorially and cinematically familiar).

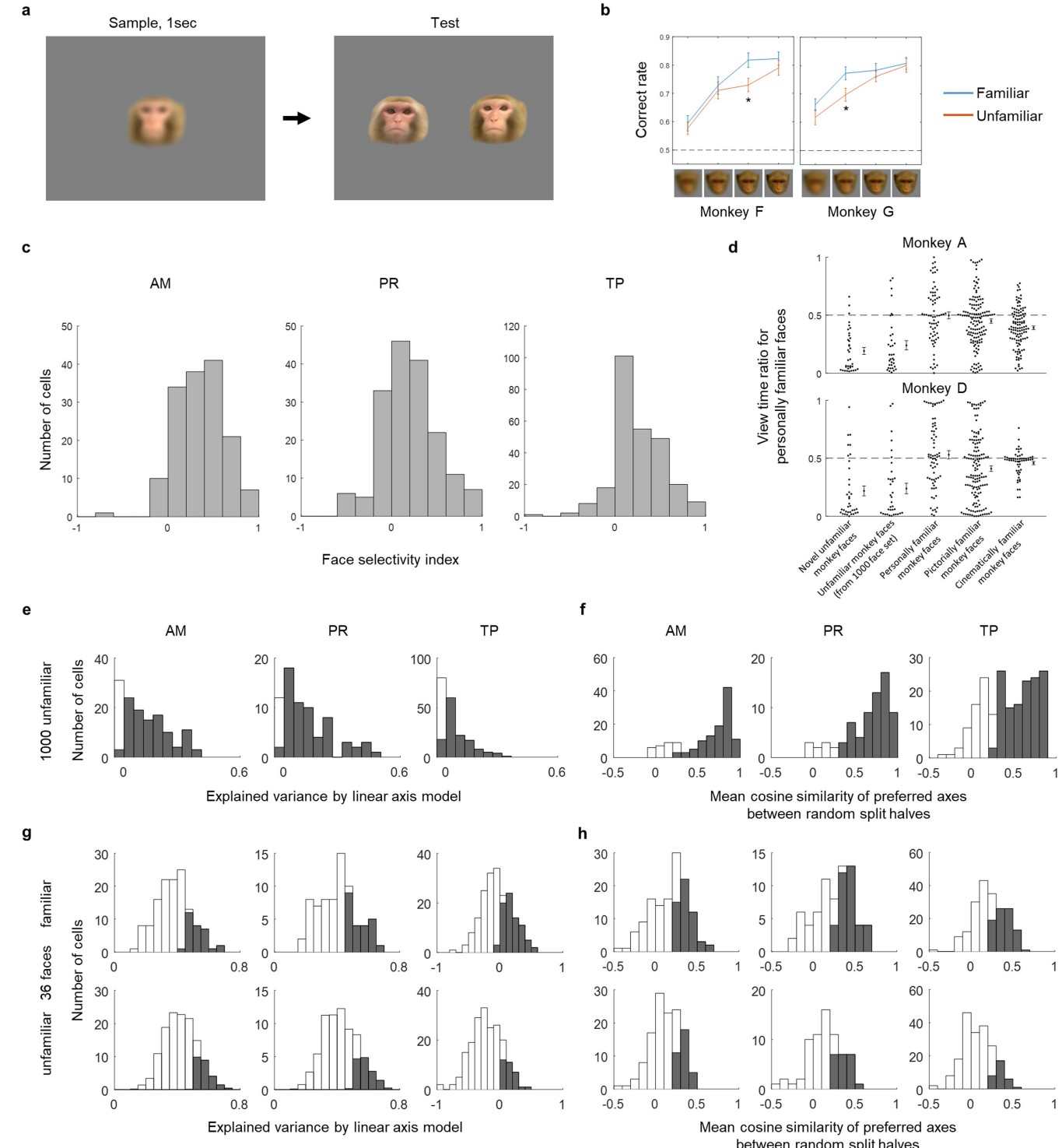

**Extended Data Fig. 3 | Quantification of familiarity-related behavior, face selectivity, and axis tuning. a**, Schematic illustration of face identification task, a sample face with different Gaussian blur level was presented for 1 s followed by a test period with two faces presented side by side. The subject had to choose the one matching the sample to get reward (see Supplementary Methods). **b**, Rate of correct performance on the face identification task across different difficulty levels (accomplished by varying Gaussian blur of the sample face, see Supplementary Methods); $n = 30$ faces. Error bar, SEM. **c**, Histograms of face selectivity indices computed using screening stimuli (see Supplementary Methods). **d**, Preferential looking test. Comparing looking time to personally

familiar faces versus novel unfamiliar faces, unfamiliar faces (from 1000 face set), personally familiar faces (two distinct personally familiar faces were presented on each trial), pictorially familiar faces, and cinematically familiar faces. Error bar, SEM. **e**, Distribution of explained variance by the linear axis model for responses to 1000 unfamiliar faces; shaded bars indicate the subset of cells for which the explained variance was significantly higher than for stimulus-shuffled data (1000 repeats). **f**, Distributions of mean cosine similarity of preferred axes across repeated split halves (100 repeats) of responses to 1000 unfamiliar faces for AM and PR. Same conventions as in **e**. **g**, **h**, Same as **e** and **f** but for 36 familiar and unfamiliar faces.

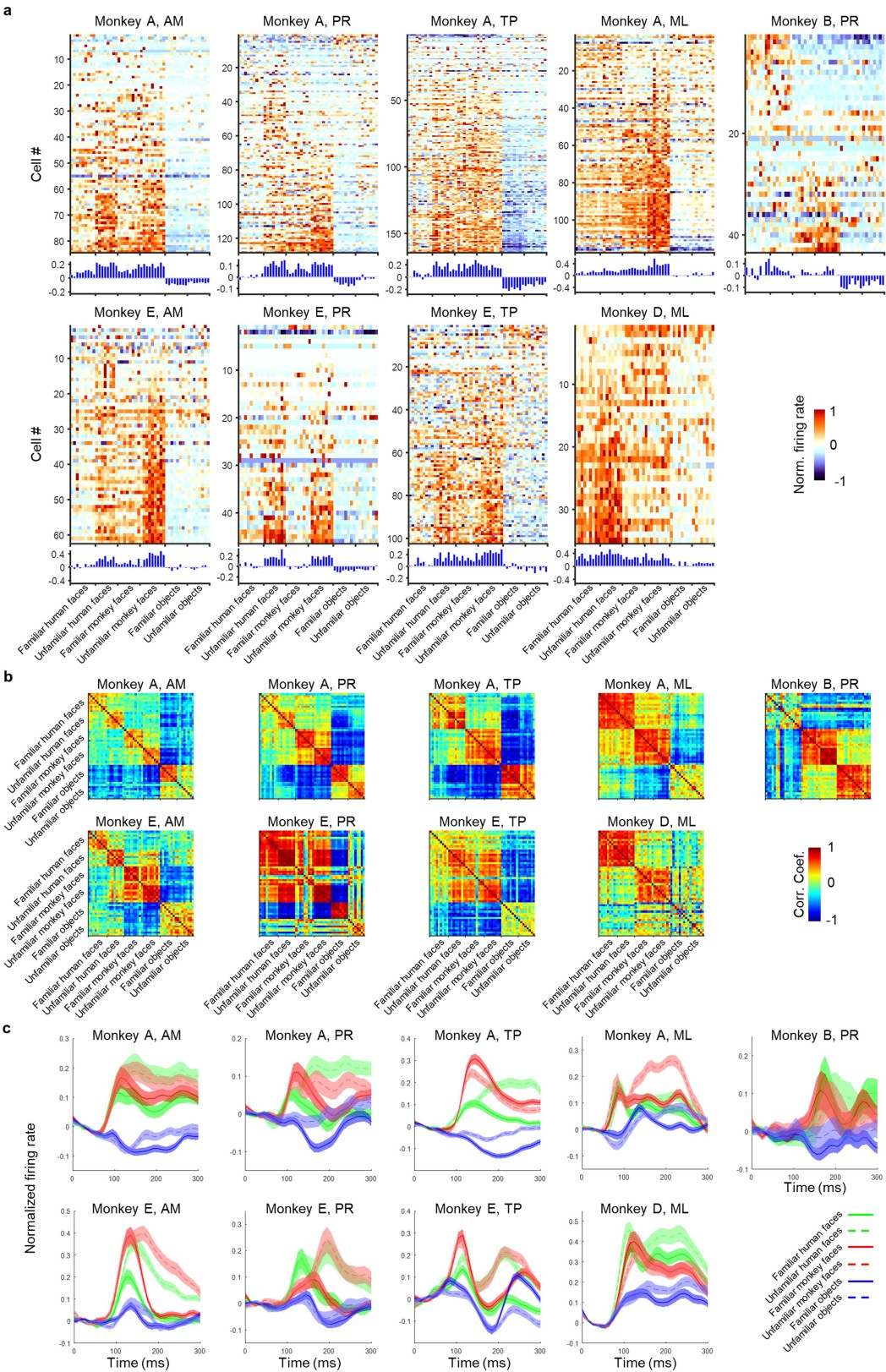

**Extended Data Fig. 4 | Main results of experiment 1 computed separately for each animal individually. a**, Responses of cells to stimuli from six stimulus categories (same as Fig. 1c). Note that Monkey C was not presented with this stimulus. Number of cells: Monkey A, AM, 84, PR, 128, TP, 164, ML, 135; Monkey B, PR, 43; Monkey E, AM, 62, PR, 46, TP, 102; Monkey D, ML, 35. **b**, Similarity (Pearson correlation coefficient) matrix of population responses for full response window (same as Fig. 1e). Number of cells same as **a. c**, Response time course averaged across cells and exemplars within each screening category (same as Fig. 1g, right). Shaded area, SEM. Number of cells same as **a**.

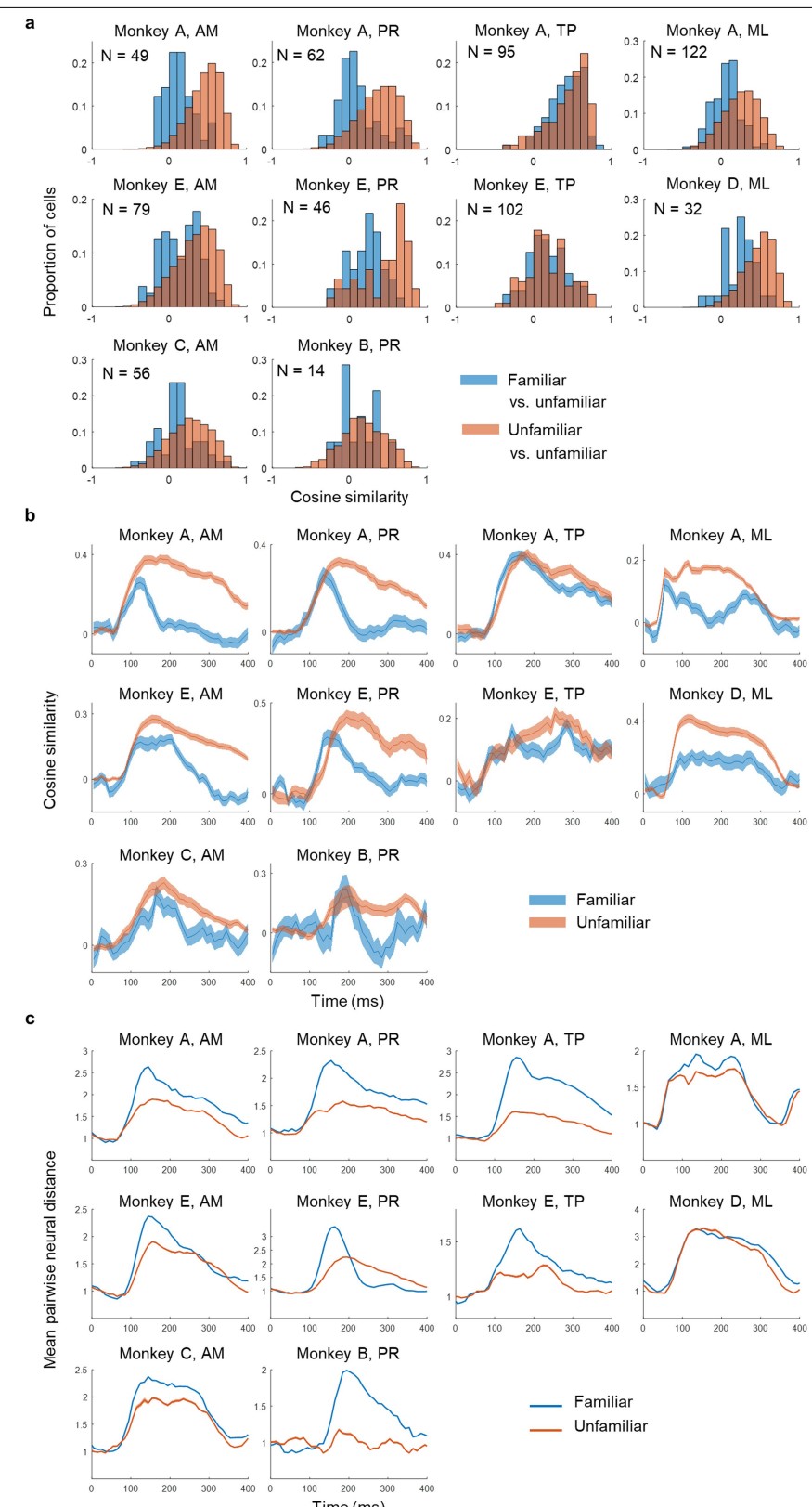

**Extended Data Fig. 5 | Main results of experiment 2 computed separately for each animal individually. a**, Population analysis comparing preferred axes for familiar versus unfamiliar faces (same as Fig. 2b). Number of cells: Monkey A, AM, 49, PR, 62, TP, 95, ML, 122; Monkey E, AM, 79, PR, 46, TP, 102; Monkey D, ML, 32; Monkey C, AM, 56; Monkey B, PR, 14. **b**, Time course of the similarity between preferred axes for unfamiliar-unfamiliar (orange) and unfamiliar-familiar (blue) faces (same as Fig. 2c). Shaded area, SEM. Number of cells same as **a**. **c**, Time course of mean pairwise neural distance (Euclidean distance between population responses) between feature-matched familiar or unfamiliar faces. Number of cells same as **a**.

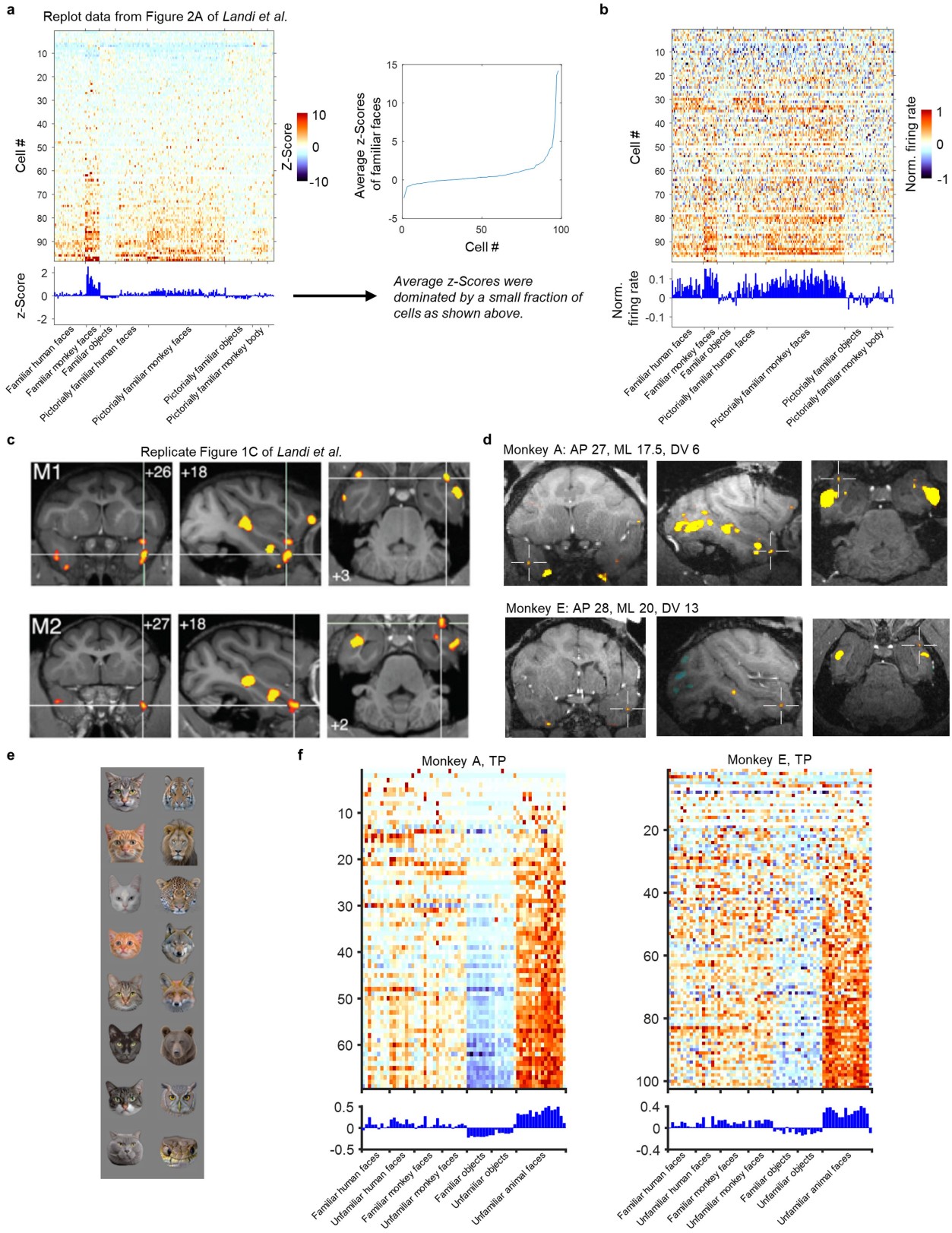

**a** Replot data from Figure 2A of *Landi et al.*

Average z-Scores were dominated by a small fraction of cells as shown above.

**c** Replicate Figure 1C of *Landi et al.*

**d** Monkey A: AP 27, ML 17.5, DV 6

Monkey E: AP 28, ML 20, DV 13

**f** Monkey A, TP

Monkey E, TP

**Extended Data Fig. 6** | See next page for caption.

**Extended Data Fig. 6 | Temporal pole face patch (TP) did not respond specifically to personally familiar faces. a**, Left: replicate of Fig. 2a from Landi et al.[23] using the data they published. Right: average z-Scores of familiar monkey faces for each cell, showing the population average of z-Scores (bar plot on the left bottom) was dominated by a small fraction of cells. **b**, Replotted population summary balancing the contribution of each cell by normalizing each cell's response by its maximum across all stimuli. **c**, replicate of Fig. 1c from Landi et al.[23] showing face patch TP in two animals **d**, MRI image overlaid with face patches showing location of TP which we recorded from in two animals. **e**, Stimuli depicting unfamiliar faces from other species; the images shown are synthetic images similar to the actual stimuli, due to difficulty in obtaining permission for publication. **f**, Responses of cells to stimuli from seven stimulus categories (familiar human faces, unfamiliar human faces, familiar monkey faces, unfamiliar monkey faces, familiar objects, unfamiliar objects, and unfamiliar faces from other species) recorded from face patch TP in two animals. Responses were averaged between 50 to 300 ms after stimulus onset.

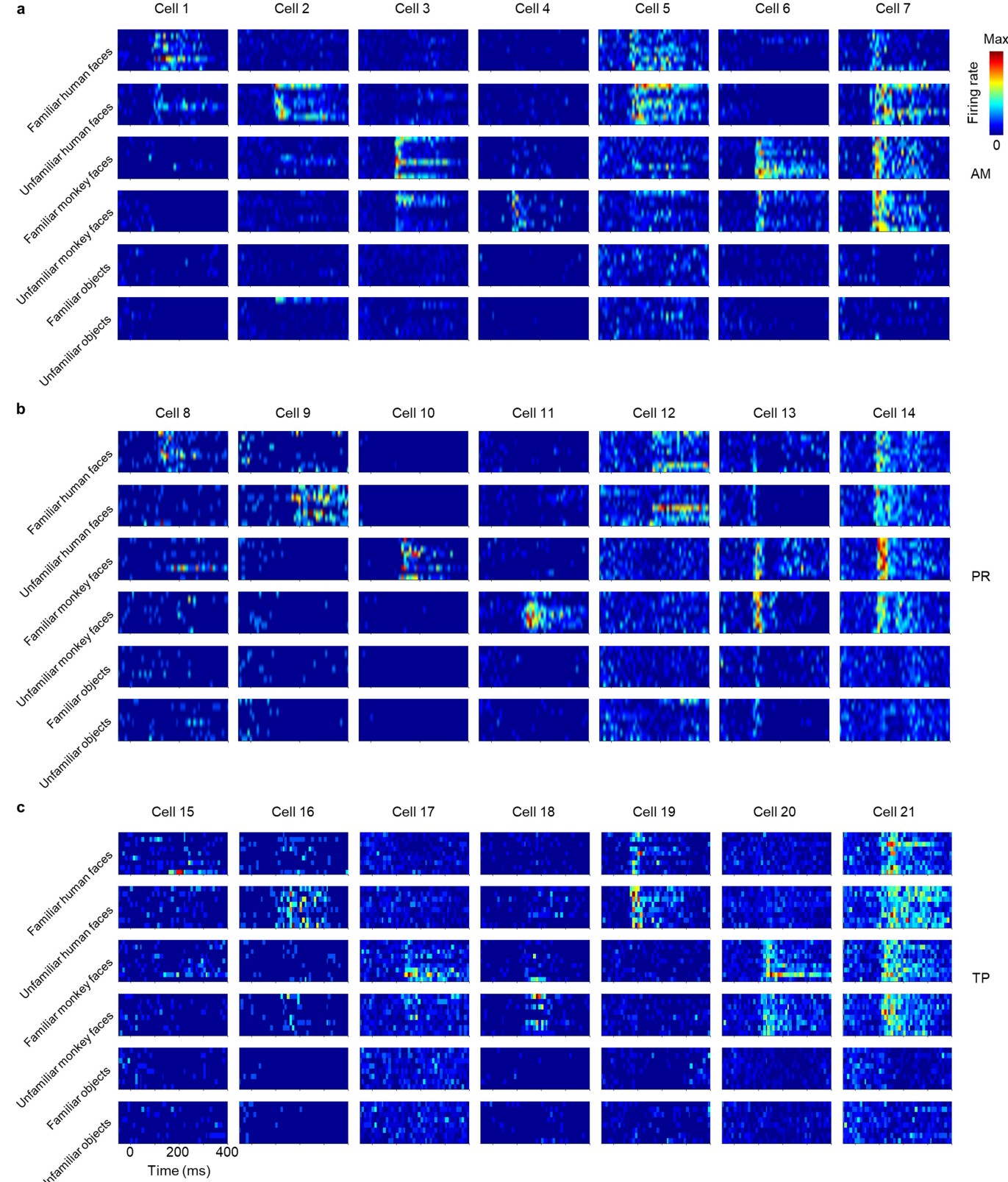

**Extended Data Fig. 7 | Responses of example neurons to familiar and unfamiliar screening stimuli. a**, Seven example cells from AM. **b**, Seven example cells from PR. **c**, Seven example cells from TP.

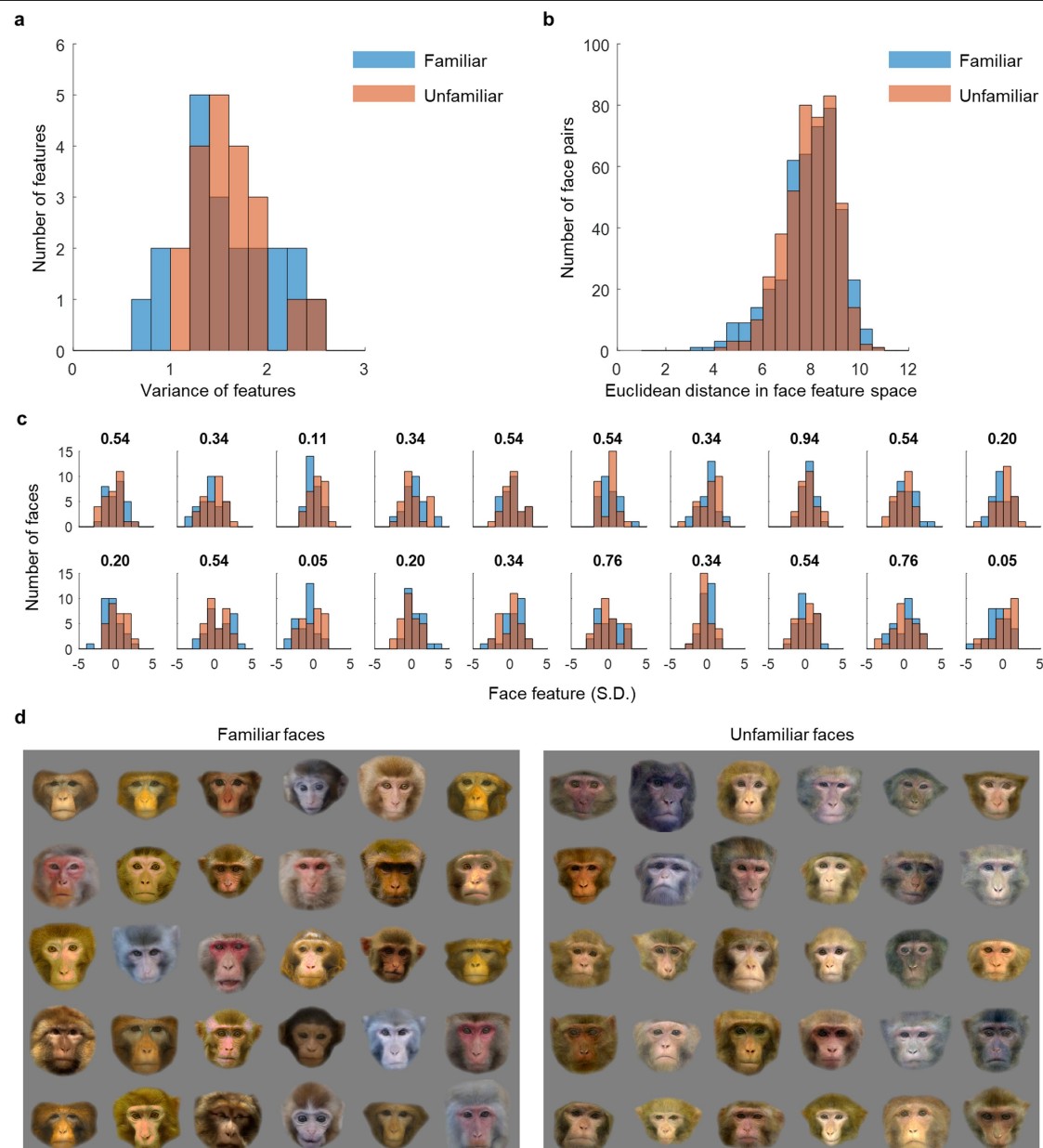

**Extended Data Fig. 8 | Matching the face features of familiar and unfamiliar faces. a**, Distribution of variances of first 20 features for 30 familiar and 30 unfamiliar feature-matched faces (two-sided Kolmogorov–Smirnov (K-S) test, $p = 0.96$, K-S statistic ($D$) = 0.15, $n$ = 20 features). **b**, Distribution of pairwise distances in face feature space (first 20 features) for the 30 familiar and 30 unfamiliar feature-matched faces (K-S test, $p = 0.51$, $D = 0.055$, $n$ = 435 face pairs). **c**, Distribution of values for the top 20 features for the 30 familiar and 30 unfamiliar feature-matched faces; the number above each plot gives the $p$ value of K-S test ($n$ = 30 faces) between the two feature distributions. **d**, Images of the 30 familiar and 30 unfamiliar feature-matched faces.

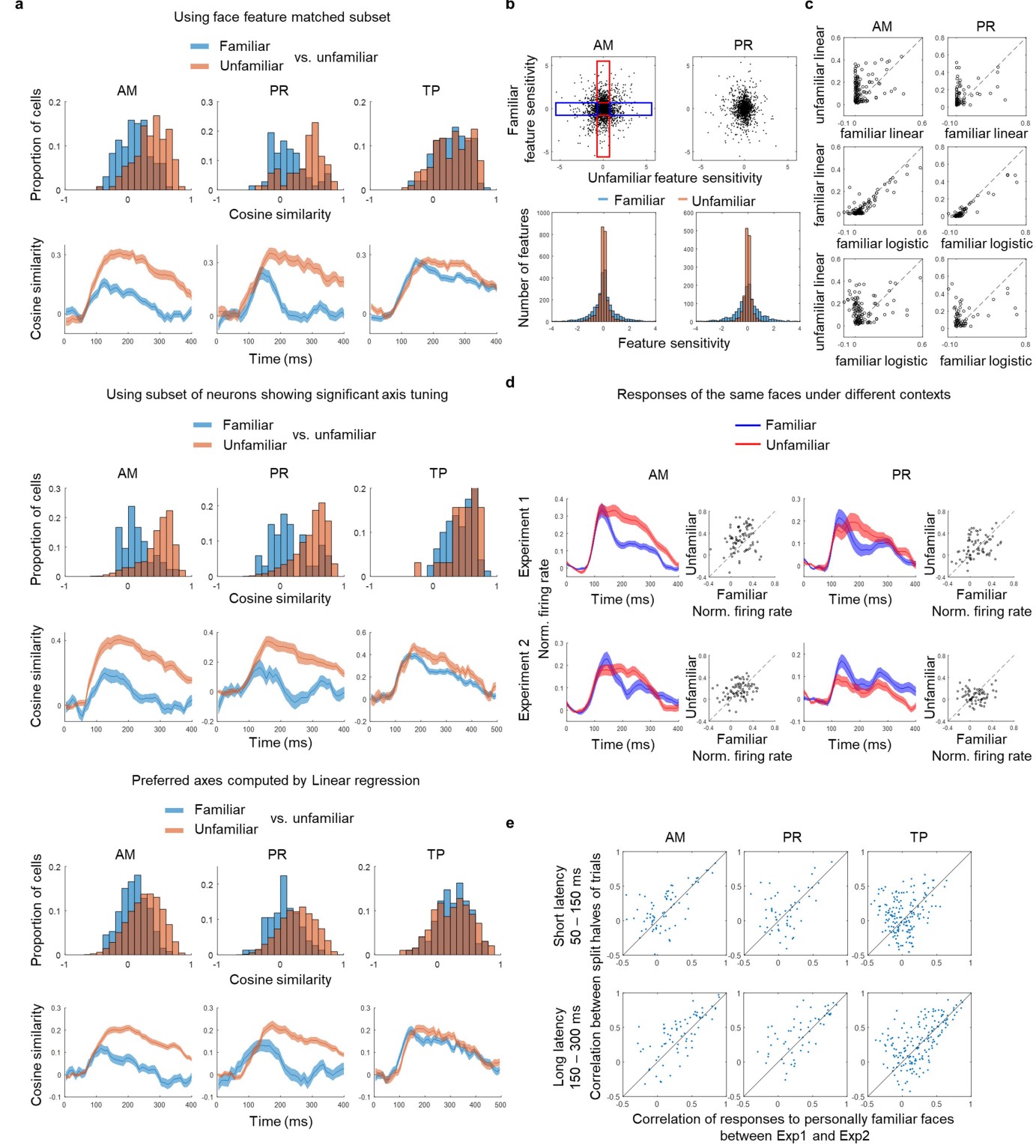

**Extended Data Fig. 9** | See next page for caption.

**Extended Data Fig. 9 | Control analyses confirming axis robustness. a**, Top, Row 1: population analysis of preferred axes for familiar versus unfamiliar faces; same conventions as in Fig. 2b except 30 familiar and 30 unfamiliar feature-matched faces were used (see Methods and Extended Data Fig. 8). Row 2: time course from the same analysis; same conventions as in Fig. 2c. Shaded area, SEM. Note that new feature-matched 36 familiar and 36 unfamiliar faces were used for TP, thus the result shown in Fig. 2c for TP is already perfectly feature matched, and is replicated here for comparison. Middle, same analysis as in Fig 2b, c except a subset of neurons showing significant axis tuning were used. Shaded area, SEM. Bottom, same analysis as Fig. 2b,c except the preferred axes were computed using linear regression rather than spike-triggered averaging (see Supplementary Methods). Shaded area, SEM. **b**, Top: scatter plot of 20 feature sensitivities (see Supplementary Methods) from 134 AM cells and 72 PR cells, for familiar (y-axis) and unfamiliar (x-axis) faces. The dots in the blue rectangles (corralling points for which sensitivity to the familiar feature goes to ~0) indicate loss of tuning for familiar faces in some cells, while the dots in the red rectangles indicate gain of tuning. Bottom: Distribution of feature sensitivity values for familiar and unfamiliar faces. This shows that on average, sensitivity for familiar faces was larger than that for unfamiliar faces. **c**, Top: explained variance for responses to 36 unfamiliar (y-axis) or 36 familiar (x-axis) faces using unfamiliar axis (fitted on 1000 - 36 faces) with linear output function (each dot is one cell, $n = 134$ cells for AM and $n = 72$ cells for PR). Middle: explained variance for responses to 36 familiar faces using unfamiliar axis with linear output function (y-axis) or a logistic output nonlinearity (x-axis); the latter values are only slightly higher. Bottom: explained variance for responses to 36 unfamiliar faces using unfamiliar axis with linear output function (y-axis) or 36 familiar faces using axis model with a logistic output nonlinearity (x-axis). The slight increase in explained variance obtained by applying a logistic output nonlinearity cannot undo the decrease caused by axis change (however, explained variance is similar using familiar axes for familiar responses and unfamiliar axes for unfamiliar responses, Extended Data Fig. 3g). **d**, Comparison of average response time courses in AM and PR to the exact same set of familiar and unfamiliar stimuli, presented in two different temporal contexts. Scatter plot: average over time window [100 300] ms (AM, N = 80 cells; PR, N = 70 cells). Top: Responses to 9 personally familiar and 8 unfamiliar monkey faces presented as part of screening stimulus (experiment 1). Bottom: responses to the same set of stimuli presented as part of thousand face stimulus (experiment 2). Shaded area, SEM. **e**, Correlation in rank order (Spearman correlation) of neuronal responses to personally familiar face stimuli at short or long latency between split halves of trials (y-axis, correlation values averaged across experiments 1 and 2) is plotted against correlation between rank order of the same faces between experiments 1 and 2; each dot represents one cell (AM, N = 80 cells; PR, N = 70 cells; TP, N = 197 cells).

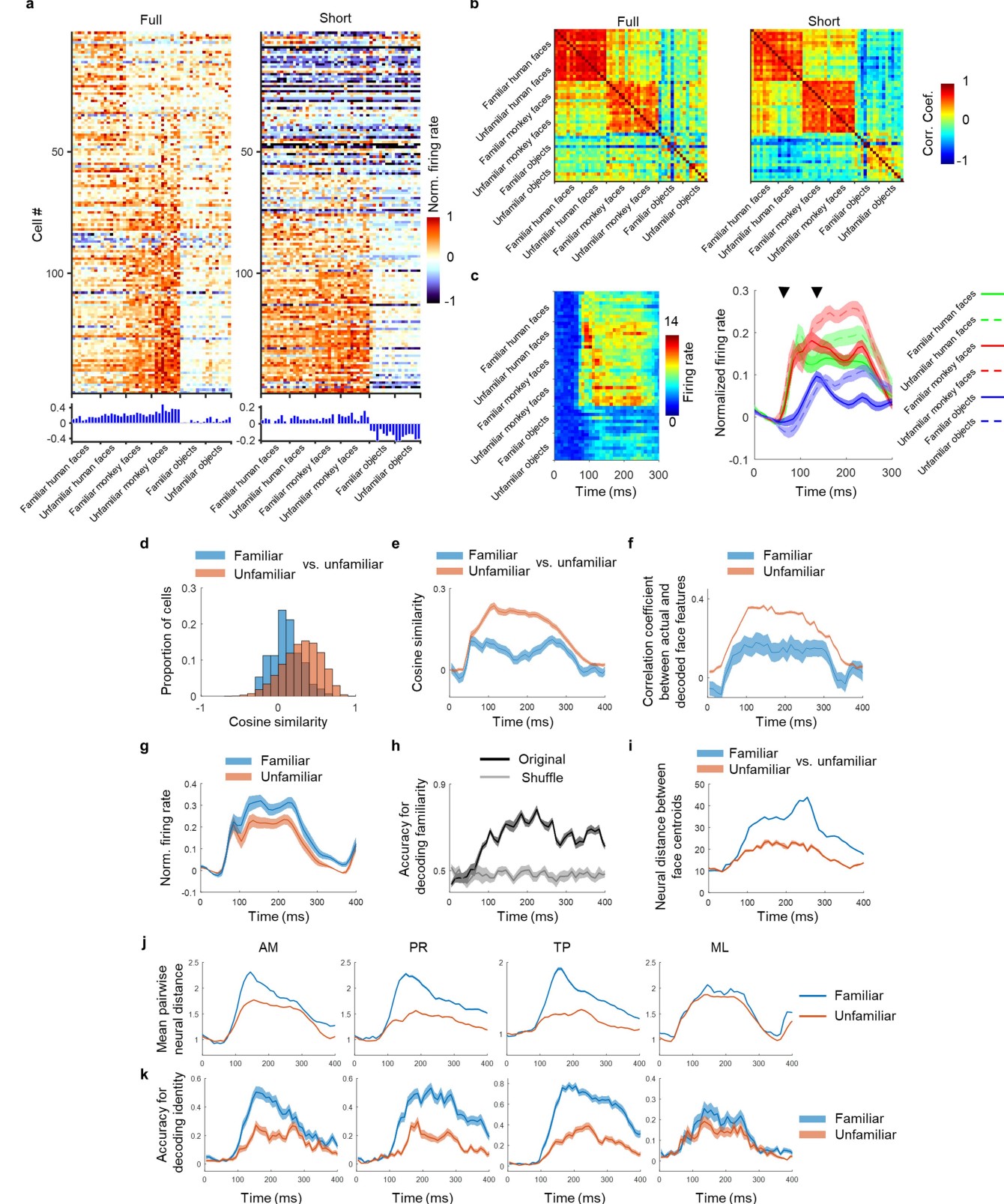

**Extended Data Fig. 10** | See next page for caption.

**Extended Data Fig. 10 | Representation of familiar stimuli in face patch ML and additional analysis of repetition suppression-related signals.**
**a**, Responses of cells to screening stimuli from six stimulus categories (familiar human faces, unfamiliar human faces, familiar monkey faces, unfamiliar monkey faces, familiar objects, and unfamiliar objects), recorded the face patch ML. Left, responses were averaged between 50 to 300 ms after stimulus onset ("full" response window). Right, same for a "short" window 50 to 125 ms. **b**, Similarity matrix of population responses for full response window (left) and short response window (right). **c**, Left: Average response time course across the ML population to each of the screening stimuli. Right: Response time course averaged across cells and category exemplars. Shaded area, SEM. Earlier arrow indicates the mean time when visual responses to faces became significantly higher than baseline (77.5 ms). Later arrow indicates the mean time when responses to familiar versus unfamiliar faces became significantly different (175 ms and 145 ms for human and monkey faces, respectively).

Responses also diverged briefly at very short latency (95 ms and 105 ms for human and monkey faces, respectively). **d**, Population analysis comparing preferred axes for familiar versus unfamiliar faces. Same conventions as Fig. 2b. **e-i**, Same analyses for the ML population ($n$ = 154 cells) as in Fig. 2c, d, Fig. 3b–d. **j**, Time course of mean pairwise neural distance (Euclidean distance) between familiar or unfamiliar faces computed using a 50 ms sliding time window, step size 10 ms, normalized by mean baseline (0–50 ms) distance between unfamiliar faces. Distances were computed using a subset of familiar and unfamiliar feature-matched faces (see Extended Data Fig. 8). **k**, Time course of face identity decoding accuracy for 30 familiar (blue) or unfamiliar (orange) feature-matched faces, computed using a 50 ms sliding time window, step size 10 ms. Shaded area, SEM. Half the trials were used to train a linear classifier and decoding performance was tested on the remaining half of trials; chance performance was 1/30.

# Reporting Summary

## Statistics

For all statistical analyses, confirm that the following items are present in the figure legend, table legend, main text, or Methods section.

| n/a | Confirmed | |
|---|---|---|
| ☐ | ☒ | The exact sample size (*n*) for each experimental group/condition, given as a discrete number and unit of measurement |
| ☐ | ☒ | A statement on whether measurements were taken from distinct samples or whether the same sample was measured repeatedly |
| ☐ | ☒ | The statistical test(s) used AND whether they are one- or two-sided<br>*Only common tests should be described solely by name; describe more complex techniques in the Methods section.* |
| ☒ | ☐ | A description of all covariates tested |
| ☒ | ☐ | A description of any assumptions or corrections, such as tests of normality and adjustment for multiple comparisons |
| ☐ | ☒ | A full description of the statistical parameters including central tendency (e.g. means) or other basic estimates (e.g. regression coefficient) AND variation (e.g. standard deviation) or associated estimates of uncertainty (e.g. confidence intervals) |
| ☐ | ☒ | For null hypothesis testing, the test statistic (e.g. *F*, *t*, *r*) with confidence intervals, effect sizes, degrees of freedom and *P* value noted<br>*Give P values as exact values whenever suitable.* |
| ☒ | ☐ | For Bayesian analysis, information on the choice of priors and Markov chain Monte Carlo settings |
| ☒ | ☐ | For hierarchical and complex designs, identification of the appropriate level for tests and full reporting of outcomes |
| ☐ | ☒ | Estimates of effect sizes (e.g. Cohen's *d*, Pearson's *r*), indicating how they were calculated |

*Our web collection on statistics for biologists contains articles on many of the points above.*

## Software and code

Policy information about availability of computer code

| Data collection | Electrophysiological signals were collected by Plexon system. Gaze position was monitored using ISCAN system. All other experimental parameters were recorded using a custom software Kofiko (https://github.com/shayo/Kofiko) (version: Feb/4/2014). The electrode trajectories were planned using custom software (https://github.com/shayo/Planner) (Revision 93 Feb/19/2014). |
|---|---|
| Data analysis | Functional imaging data are prcoessed with Freesurfer and FSL. Multichannel recorded electrophysiological data was automatically sorted by Kilosort2 (github.com/MouseLand/Kilosort2) and manually refined in Phy (github.com/cortex-lab/phy). Custom code written in MATLAB was used for analysis. The code that reproduce the core results are available at https://doi.org/10.5281/zenodo.10460607. All other code is available from the lead corresponding author upon reasonable request. |

For manuscripts utilizing custom algorithms or software that are central to the research but not yet described in published literature, software must be made available to editors and reviewers. We strongly encourage code deposition in a community repository (e.g. GitHub). See the Nature Portfolio guidelines for submitting code & software for further information.

## Data

Policy information about availability of data

 All manuscripts must include a data availability statement. This statement should provide the following information, where applicable:
- Accession codes, unique identifiers, or web links for publicly available datasets
- A description of any restrictions on data availability
- For clinical datasets or third party data, please ensure that the statement adheres to our policy

> The data set of neural responses to thousand monkey face set are available at https://doi.org/10.5281/zenodo.10460607.
> PrimFace database: http://visiome.neuroinf.jp/primface
> FERET database: https://www.nist.gov/itl/products-and-services/color-feret-database
> CVL Face Database: http://www.lrv.fri.uni-lj.si/facedb.html
> MR2 face database: https://osf.io/skbq2/
> Chicago Face Database: https://www.chicagofaces.org/
> CelebA CelebFaces Attributes Dataset: https://mmlab.ie.cuhk.edu.hk/projects/CelebA.html
> FEI Face Database: fei.edu.br/~cet/facedatabase.html
> PICS Psychological Image Collection at Stirling: pics.stir.ac.uk
> Caltech faces 1999: https://data.caltech.edu/records/6rjah-hdv18
> Essex Face Recognition Data: http://cswww.essex.ac.uk/mv/allfaces/faces95.html
> The MUCT Face Database: www.milbo.org/muct
> All other data are available from the lead corresponding author upon reasonable request.

## Research involving human participants, their data, or biological material

Policy information about studies with human participants or human data. See also policy information about sex, gender (identity/presentation), and sexual orientation and race, ethnicity and racism.

| | |
|---|---|
| Reporting on sex and gender | N/A |
| Reporting on race, ethnicity, or other socially relevant groupings | N/A |
| Population characteristics | N/A |
| Recruitment | N/A |
| Ethics oversight | N/A |

Note that full information on the approval of the study protocol must also be provided in the manuscript.

# Field-specific reporting

Please select the one below that is the best fit for your research. If you are not sure, read the appropriate sections before making your selection.

☒ Life sciences          ☐ Behavioural & social sciences          ☐ Ecological, evolutionary & environmental sciences

For a reference copy of the document with all sections, see nature.com/documents/nr-reporting-summary-flat.pdf

# Life sciences study design

All studies must disclose on these points even when the disclosure is negative.

| | |
|---|---|
| Sample size | Sample sizes were chosen in a manner commensurate with similar previous studies. |
| Data exclusions | We recorded from every neuron encountered. Only visual responsive units were considered for further analysis. |
| Replication | Results were replicated across 2-3 different animals for each experiment independently. |
| Randomization | The visual stimuli were shown in a random order. Organisms random allocation is not relevant to this study, different subjects were used to repeat the same experimental condition. |
| Blinding | Investigators were not blinded to experimental groups due to the nature of the experiments. |

# Reporting for specific materials, systems and methods

We require information from authors about some types of materials, experimental systems and methods used in many studies. Here, indicate whether each material, system or method listed is relevant to your study. If you are not sure if a list item applies to your research, read the appropriate section before selecting a response.

## Materials & experimental systems

| n/a | Involved in the study |
|-----|----------------------|
| ☒ | ☐ Antibodies |
| ☒ | ☐ Eukaryotic cell lines |
| ☒ | ☐ Palaeontology and archaeology |
| ☐ | ☒ Animals and other organisms |
| ☒ | ☐ Clinical data |
| ☒ | ☐ Dual use research of concern |
| ☒ | ☐ Plants |

## Methods

| n/a | Involved in the study |
|-----|----------------------|
| ☒ | ☐ ChIP-seq |
| ☒ | ☐ Flow cytometry |
| ☐ | ☒ MRI-based neuroimaging |

# Animals and other research organisms

Policy information about studies involving animals; ARRIVE guidelines recommended for reporting animal research, and Sex and Gender in Research

| Laboratory animals | Seven male rhesus macaques (Macaca mulatta) of 5-13 years old were used in this study. |
|---|---|
| Wild animals | The study did not involve wild animals. |
| Reporting on sex | This study was conducted using only male animals. |
| Field-collected samples | The study did not involve field-collected samples. |
| Ethics oversight | All procedures conformed to local and US National Institutes of Health guidelines, including the US National Institutes of Health Guide for Care and Use of Laboratory Animals. All experiments were performed with the approval of the Caltech and UC Berkeley Institutional Animal Care and Use Committee. |

Note that full information on the approval of the study protocol must also be provided in the manuscript.

# Magnetic resonance imaging

## Experimental design

| Design type | Block design |
|---|---|
| Design specifications | Each block lasted 24 s blocks (each image lasted 500 ms). In each run, the face block was repeated four times and each of the non-face blocks was shown once. A block of grid-scrambled noise patterns was presented between each stimulus block and at the beginning and end of each run. Each scan lasted 408 seconds. |
| Behavioral performance measures | Subjects' eye position was monitored using an infrared eye tracking system (ISCAN). Juice reward was delivered every 2-4 s in exchange for maintaining fixation on a small spot (0.2 degree) |

## Acquisition

| Imaging type(s) | Functional and anatomical imaging |
|---|---|
| Field strength | 3 Tesla |
| Sequence & imaging parameters | T1-weighted anatomical volumes were measured with MP-RAGE sequence( TR 2,300 ms; IR 1,100 ms; TE 3.37 ms; 0.5 mm isotropic voxels) . EPI volumes were acquired in an AC88 gradient insert (Siemens) TR was 2000 ms,TE was 17 ms, voxels were 1 × 1 × 1 mm with an no gap between slices. Matrix size was (96, 96, 64) (read [x], phase [y], slice [z]), the field of view was 96 × 96 mm in-plane. Flip angle was 80° |
| Area of acquisition | Whole brain |
| Diffusion MRI | ☐ Used   ☒ Not used |

## Preprocessing

| Preprocessing software | Analysis of functional volumes was performed using the FreeSurfer Functional Analysis Stream (Massachusetts General Hospital). Volumes were corrected for motion and undistorted based on acquired field map. |
|---|---|
| Normalization | No normalization needed as analysis only compare data from the same scan. |

| Normalization template | We did not normalize any imaging data into template. All the analysis were done in the single subject's original space. |
| Noise and artifact removal | We remove the linear or quadratic trends in the timeseries. |
| Volume censoring | Motion noises were removed by putting the motion parameters as the regressors in the GLM analysis. |

## Statistical modeling & inference

| Model type and settings | The analysis used only first-level analysis. |
| Effect(s) tested | We ran t-tests between different conditions within each single subject. |

Specify type of analysis: ☒ Whole brain   ☐ ROI-based   ☐ Both

| Statistic type for inference | All the analyses were done using voxel-wise inference. |

(See Eklund et al. 2016)

| Correction | We did not apply any multiple-comparison correction in the fMRI imaging analysis. |

## Models & analysis

| n/a | Involved in the study |
| --- | --- |
| ☒ ☐ | Functional and/or effective connectivity |
| ☒ ☐ | Graph analysis |
| ☒ ☐ | Multivariate modeling or predictive analysis |

