## [Peer Review File · Nature]

Manuscript Title: Temporal multiplexing of perception and memory codes in IT cortex

Reviewer Comments & Author Rebuttals

Reviewer Reports on the Initial Version:

Referees' comments:

Referee #1 (Remarks to the Author):

This is a tour de force experiment and analysis addressing the fascinating question of how face representation, based on neural axis tuning in a face shape/appearance space, as previously shown by this group, changes between unfamiliar faces (previously studied) and familiar faces (until now not studied). The results clearly show that the tuning axis model fit to a given neuron's responses to 1000 unfamiliar faces explains the early response phase for 36 familiar faces in anterior face patch AM and in a recently discovered perirhinal face patch (PR), but later responses diverge substantially.

The authors characterize this change as a shift in axis direction in the face parameter-based space. But the shifted axis explains a significant amount of variance for only a fraction of neurons (extended data Fig. 9e). The small set of 36 familiar face is not adequate for constraining the 20 parameter model, as the authors recognize explicitly. And the fact that explained variance goes up nonetheless suggest that the new axis model is overfitting the small number of data points. It is clear that a greater axis shift is required to fit/overfit familiar faces compared to 36 unfamiliar faces. But, as discussed below, the alterations in neural responses are potentially explainable with something simpler than the 20 coefficient changes used here. The authors conclude that familiar faces induce a delayed shift in tuning axis direction in AM and PR, while at the same time admitting that they cannot rule out a different explanation in terms of nonlinear tuning changes. That is a disappointingly tentative conclusion.

But I think the authors could extract much more definite results from this magnificent dataset. The first alternative to explore is nonlinearity, which the authors mention frequently but believe cannot be tested here. (They also seem to say that nonlinearity could only explain the residual variance after axis fitting, but axis overfitting could capture nonlinearities.) But a general form of nonlinearity could be implemented here by just passing the dot product from the original 1000 unfamiliar face-based model through a final, fitted logistic function. If nonlinearity emerges, then the shape of the logistic function should change across time, from basically linear to more of a step function. Delayed nonlinearity is a simple change, describable with simple models, and already known to occur in IT on a delayed timeframe as observed here, following initially linear responses, also as observed here (Brincat & Connor, Neuron 2006). This would be mechanistically far simpler than changing every neuron's tuning direction in the face space. And, nonlinear interactions between features/parts are known to increase with stimulus exposure in IT (Baker, Behrmann, & Olson, Nature Neuroscience 2002). The sparsely responding neurons mentioned in the text might also be captured by a face space axis combined with a nonlinear logistic function; they might actually represent one end of a continuum of nonlinearity in the AM and PR populations. The logistic function cannot capture the

full complexity of potential nonlinearities in the 20 dimensional space, but it can capture the overall supra- or sub-additivity they may produce.

Another hypothesis much simpler than changing tuning in 20D space is that neurons become less sensitive to features that are not diagnostic for the small group of individuals they know. This could be modeled easily by deleting axes in the original tuning space to search for a lower dimensionality in which a given neuron operates to discriminate just 36 individuals. This would avoid fitting new values, and it would again have a known type of neural implementation, similar (or perhaps identical) to attentional suppression of distractor information. It would also explain the lower average responsiveness to familiar faces under at least some conditions.

Even if changing direction in the space is essential, this could also be constrained by using some metric to minimize the number of coefficients that are changed. Just as distractor responses can be suppressed by attention, target responses can be magnified. Each dimension/coefficient in these analyses corresponds to some combined set of measurable changes in face appearance. Some of those differences might be more diagnostic for the set of 36 familiar individuals. Changes in just a few coefficients could be adequately constrained by 36 data points, avoiding overfitting, and thus potentially achieving more significance in the randomization test.

These analyses, alone or in combination, could also produce results that are clearly interpretable. For a given neuron, the few dimensions that best capture the data could be cashed out as visualized changes in face shape/appearance. Those dimensions, across neurons, could tell us about what specific feature combinations are used to identify familiar faces, and whether those specific features are particularly diagnostic for the individual faces in the familiar group. That would add a lot to the scientific interest of this paper. At the least, such analyses could rule out alternatives to the authors' favored hypothesis of tuning direction changes in 20D space.

Ed Connor

Referee #2 (Remarks to the Author):

She et al. conducted a study to investigate how responses in macaque AM face patch and perirhinal cortex differ for familiar and unfamiliar faces. The main finding is that familiar faces were differently represented than unfamiliar faces at different latencies. The results presented in this study will be of interest to researchers studying face and object recognition. What impresses me about the manuscript is the thoroughness with which the research was done and the presentation of both raw data and higher-level summaries.

However, I see several issues with this manuscript that I would invite the authors to address.

- The main finding is that familiar and unfamiliar faces are represented at different times in two brain regions. However, there is a potential flaw in this interpretation. In figure 2e there seems to be a difference in contrast between familiar and nonfamiliar faces. Unfamiliar faces seem paler than familiar faces (extended data figure 2) for both human and monkey faces. The authors say that they compared the similarity of features in their feature model, however, this is not a standard model used to evaluate features. I would like to invite the authors to check how features of familiar and unfamiliar images differ in some of the standard low-level feature models like GIST, HMAX, or early layers of a deep convolutional neural network (be it AlexNet or other architecture).

- When the authors mention that they defined 30 familiar and unfamiliar faces that were feature-matched, they should describe in the results section how this feature match was determined (apart from discussing it in details in the methods section).

- There is a different number of familiar and unfamiliar faces (34/16 and 36/100 in two experiments). While it is understandable that there is a limitation in the number of familiar faces that the animals were exposed to, I was wondering why the number of unfamiliar faces was not matched to that of familiar faces. It would be helpful if the authors could address that.

- It is hard to follow the results section where the authors discuss the axis code without diving into the methods. There should be more explanation of what axis code is in the results section so that the reader can interpret the data.

- In the discussion section, it is mentioned that a simple computational model could recapitulate the study's findings but this finding is not described in the results section, so please describe it if this result is to be discussed.

- In the discussion section, the authors state that their results contradict some of the current literature but they do not specify where these differences may have come from. In general, the discussion section would benefit from more details on the relevant literature and its relation to the current study.

- The study would benefit from more discussion about the effects of familiarity and the time course of representations. For example, Dobs et al. 2019 has studied the effects of familiarity in the MEG responses in humans but this study is not cited.

Minor points:

- The figures have too many subpanels (e.g., labelled e1, e2, e3, e4). I would label them with successive letters rather than a mix of letters and numbers.
- In extended data figure 1., the colour bar should be labelled in the figure. It is not clear why the colorbar is not within the same range for all monkeys.
- It would be easier for reviewers if the authors embed figures in the text for the potential revision.

Referee #3 (Remarks to the Author):

Many studies have shown that neurons in the inferotemporal (IT) cortex represent the sensory percept of visual objects, more specifically, neurons clustered in face patches within IT have been shown to represent visual faces. But do these same neural populations represent the memory of visual objects or faces? In this manuscript the authors tackle this question by focusing on long-term memory, and recording from three face patches, two in IT (face patch AM and ML) and one face patch in perirhinal cortex (PR). To investigate how long-term memory might be represented in these face patches they compare neural responses to images of familiar and unfamiliar objects. Personally familiar images depicted people, monkeys, and objects that the animals interacted with on a daily basis. To confirm familiarity, two of the five monkeys studied were tested behaviorally, and showed highly significant preferential looking toward unfamiliar face stimuli and away from familiar stimuli. It is important to note that no animals were behaviorally tested to see if they are better at identifying familiar faces from each other.

This paper is not lacking in results. Many of the results are interesting, although I am not sure how much interest these results will garner outside of the community that studies visual object/face recognition. In truth it would take a serious study of previous work before many of the results can even be understood. I commend the authors for attempting to fit all their results in a short manuscript. Unfortunately, this doesn't come without a heavy toll on the reader. Understanding each result requires tracking data analyses and ideas interspersed between text, sub-figures, supplemental text, and supplemental sub-figures. In tracking these results, I was left with more questions than answers and I am not convinced that they cracked the neural code for face memory. I do believe that they propose a different way of looking at encoding face memory, and given the room in a longer manuscript in a more specialized journal they might be able to present a more cohesive story that systematically goes through their different manipulations and results. A longer manuscript would allow them to more clearly explain their challenges to previous ideas and results on how familiarity impacts the neural code for objects and faces in IT.

Based on the title of the paper and the abstract, the authors argue that the novelty of their study lies in their claim that it uncovers "The neural code for face memory". In this neural code familiar faces are represented in distinct neural subspace that at short latency is shifted relative to the subspace for unfamiliar faces and then distorted in order to increase neural distances between familiar faces at long latency. In the absence of behavioral data that would confirm that monkeys are better at identifying familiar faces from each other, a more appropriate title would be "A potential neural code for long-term face memory".

The proposed neural code is within the framework of the "axis code", which is an elegant geometric approach to the neural coding of face identity described in an important previous publication by the same lab. Using this framework, the authors show that familiarity causes neural responses in AM and PR (even ML based on my reading of extended figure 20) to go off axis. The authors assert that this axis change serves to increase the neural distance between familiar faces.

The crux of the authors claim for a link between the axis change and the increase in neural distance is based on an analysis shown in figure 2h2 in which they show the mean pairwise neural distance

between 30 familiar faces is larger when using a familiar axis compared to using an unfamiliar axis. The result is very clear in figure 2h2, although I would have liked to see the same analysis done for the unfamiliar faces for comparison. I would have also preferred that the authors didn't jump from looking at 9 familiar faces in one analysis, to 36 familiar faces in another, to 30 completely different faces in yet another. All of the results are presented in one figure and assumes without verification that if the same analysis was applied to a different image set it would produce the same results. What makes this even more complicated is that some of the data were collected after the monkeys spent a month learning/watching videos of 27 faces to make them familiar.

There are two potential issues with the link between the change in axis and the increase in neural distance between familiar faces:

1-Neurons in areas AM and PR show a change in axis with a commensurate change in face pairwise neural distance. But the data is pooled across animals, 3 animals in the case of AM and 2 animals in the case of PR. I would have liked to see this analysis broken down by monkey especially since monkey B doesn't show a big change in axis.

2-Neurons in area ML(extended figure 20d), at least to my eye, shows a change in axis (blue histogram for familiar faces is shifted from the red histogram (unfamiliar faces)), without a commensurate change in neural distance between familiar faces. ML is based on data from one animal. The authors claim that "Overall, representations of familiar versus 11 unfamiliar faces were much more similar in ML compared to AM (Extended Data Fig. 20d12 k)", so I might be missing something. From a graphical standpoint the changes in cosine similarity would be much easier to evaluate and compare if they were plotted as a scattergram in order to see the axis change on a neuron by neuron basis. The histogram hides whether the axis change applies to all cells or is mostly driven by a few cells that show large changes.

Even if the change of axis does indeed increase the neural distance between familiar faces, in the absence of behavioral data that tests face identification performance for familiar and unfamiliar faces and shows that the difference in identification performance is predicted by the neural effects reported, it is premature to claim that the proposed code is "the neural code for face memory."

The authors also show that inactivation of PR using muscimol doesn't impact the change in axis-code between familiar and unfamiliar faces in AM. Inactivation of PR does change the neural responses in AM. It isn't clear whether inactivation of PR impacts the neural distance between familiar faces, or whether it changes the similarity matrix. As far as I can tell, these data were also only collected in one animal.

I agree with the authors that their results pose a challenge to repetition suppression as a model for familiarity encoding. It is quite interesting that repetition suppression depended heavily on temporal context. In the screening stimulus set where familiar monkey faces and human faces and objects were mixed with their unfamiliar counterparts, all familiar object categories show a delayed suppression. However when familiar monkey faces were presented with a 1000 unfamiliar faces, the familiar faces showed enhanced responses. In the context of short term memory, it has been shown that not only does repetition suppression depend on temporal context, but it is also heavily

modulated by task (Miller and Desimone, 1991, 1994).

But is the axis-code shift and distortion immune to such response modulation?

The authors attempt to address this by measuring responses to low-contrast faces. They show that axis-similarity was not changed by the contrast-varied faces which were meant to decrease response gain. However, the authors neglect to share with us how big the change of contrast was and how it actually impacted the responses of the neurons.

I was also struck by the analysis the authors performed in Extended figure 18. The authors focus on the fact that the difference between responses to 9 personally familiar and 8 unfamiliar faces depends on the larger stimulus set used. But what caught my attention was that the enhancement they describe in Fig 3b in their “thousand faces” experiment, seems to be largely driven by the cinematically familiar faces. These faces are the newly “learned” familiar faces. The personally familiar faces show less enhancement if any. When plotting the axis tuning example cells in Fig 2a1, I am puzzled by why they restrict themselves to the 9 personally familiar, and why they don’t include all familiar faces. I also wonder how much the changes in preferred axis is actually driven by the enhancement in responses to the cinematically familiar faces? The authors address this issue by conceding that their 36 unfamiliar faces are not necessarily white (uncorrelated in feature space). They try a “whitening control” (Extended figure 15b) which didn’t affect their results, but as they point out in the discussion, focusing on 36 familiar faces limits their ability to fully characterize the distortion in the linear subspace that they claim is produced by familiarity.

Is the axis code itself immune to stimulus presentation time and stimulus design? As the authors note in their supplemental text, Koyano et al. (2021) using different stimuli and different presentation times show that feature tuning in face cells can be V-shaped. The authors speculate that the V-shape tuning observed is due to an over-representation of the average face in the stimulus set because they continue to see ramp tuning even when they increase their stimulus duration. But the dependence of ramp tuning on stimulus set/ temporal context, makes it difficult to argue that it is more robust than repetition suppression.

In setting up their results the authors pitch two extreme schemes for representing familiar faces (fig 1a), in the first faces are represented as points on a continuous low-dimensional object space and the other is discontinuous, “sparsened”, with separate attractors representing the different objects. They argue that their results challenge the “sparsification” model. But considering that they found a small population of sparse cells in both AM and PR that respond most strongly to images of particular familiar individuals, it is very hard for me to take the challenge seriously. Furthermore, I would be interested to look at analysis of sparseness for the screening set to see if there is difference in sparseness when comparing responses to familiar versus unfamiliar objects and faces. To my eye, the responses similarity matrices in Fig 1d shows hints of changes in sparseness and it would be informative to quantify those changes to see if they are consistent with previously reported sparseness changes with familiarity (ref 15-20).

Moreover, on the topic sparseness, the authors argue that changes in sparseness (“sparsening”) can’t explain why responses to familiar faces go off-axis. Unfortunately, I found Extended Figure 8 to

be a bit opaque in making that case. A change in sparseness doesn't preclude a change in rank order. If the changes in sparseness associated with increases in familiarity include a change in which image the cells respond best, such a change would be consistent with the off-axis response described in this paper. In general I found the schematic figures (Figure 1a, Figure 3g and Extended Figure 8) to be detached and unrelated to the actual data analysis figures used to support the same ideas.

I understand the challenge of working with non-human primates. Especially considering the numerous results presented in this manuscript. I found it extremely difficult to figure out which monkey contributed to what results. Many of the conclusions are weakened by the fact that results represent data from only one monkey. As far as I can tell, the AM muscimol experiment (monkey e) and the ML recordings (monkey d) were done in only one monkey. The PR recordings were pooled across two animals, but Monkey B only contributed 14 neurons, responses. For the screening set on Monkey B (PR) it is very difficult to see whether there is a difference in average firing rate between familiar and unfamiliar faces. For the 1000 monkey face set, Monkey B (PR) shows a very large difference with cells barely responding to unfamiliar faces. Interestingly the changes in preferred axis between familiar and unfamiliar faces are less robust (Extended Figure 4). Unfortunately, no behavioral data were collected on Monkey B to see if the behavior explains the smaller differences observed. Considering the variability, I think it would be best not to pool neurons across animals.

Many figures don't have error bars. Figure 2a and Extended Figure 10 (no error bars on the familiar and unfamiliar data points). Figure 2f,g, having error bars on mean neural distance between faces would give important context to the magnitude of the difference. In extended Figure 4, it would be good to show the similarity matrices for each monkey and area.

Author Rebuttals to Initial Comments:

Reviewer responses

We thank all three reviewers for their careful reading of our paper and helpful suggestions. By addressing the concerns raised and incorporating our responses into the manuscript, we believe we have significantly improved the paper. We apologize for the long delay in turn around, due to multiple factors including the pandemic as well as relocation by the senior author to a new institution.

General summary of the revision (relevant to all reviews)

Below, we summarize the major new experiments and analyses we performed since the last version. We would like to emphasize that even though it took a lot of control experiments and analyses to get there, *our central story is a simple one*, namely, that face memory is encoded through a long-latency change in axis. This challenges a prevailing assumption about how long-term memory is represented, the repetition suppression hypothesis. We believe this finding is fundamentally important because it shows that IT cortex is not simply representing object features, but instead is also directly representing object memories. We took extreme care to ensure that the unfamiliar and familiar faces were feature matched (see point #4 below), yet we still observed this phenomenon. While most previous work on neural correlates of familiarity can be explained by simple gain changes, we believe that the axis change for familiar faces reveals a much deeper coding principle concerning how the brain represents objects and stores long term memories of them using the same neural populations.

Major new experiments and analyses:

1) New recordings in face patch TP. Since our previous submission, a paper was published reporting that a face patch in anterior IT, TP, is specialized for representing familiar faces (Landi et al., Science 2021). Because this claim clearly calls into question our central conclusion that AM and PR are the site of long-term memory storage for faces, we decided to investigate further and repeated our experiments in TP in two monkeys (Monkeys A, E). We found that contrary to the claim by Landi et al., cells in TP *are not specialized for representing familiar faces*, and instead respond even more strongly to unfamiliar faces, just like cells in AM and PR (Fig. 1c, g); moreover, the response in TP to faces of other species which our monkeys had never encountered before was even stronger than that to human or monkey faces (Extended Data Fig. 5e, f). We believe this may be due to Landi et al. not including truly unfamiliar faces in their stimulus set; instead, they showed the same screening set for each recording session, which included faces designated as unfamiliar that likely became pictorially familiar through repeated exposure. Most interestingly, *cells in TP did not show long-latency axis change*, unlike in AM and PR (Fig. 2b-d). This provides an additional control showing that the observation of axis change in AM and PR is not a stimulus-related artifact. Furthermore, it shows that long-latency axis change is a circumscribed phenomenon that can be localized to regions of the temporal lobe implicated in long-term memory by substantial independent evidence (e.g., see Miyashita, NRN 2019).

2) New behavioral experiment on familiar face recognition in two new monkeys. Reviewer 3 cited the absence of behavioral assessment of our monkeys' ability to recognize familiar versus unfamiliar faces as an important shortcoming. We therefore trained two monkeys on a face identification task that allowed us to directly compare familiar versus unfamiliar face recognition. The behavioral testing confirmed superior performance on familiar face recognition by both monkeys (Fig. 4d).

3) Additional data for muscimol experiment and PR recordings. To address Reviewer 3's concerns regarding reproducibility across animals, we conducted an additional muscimol experiment in monkey A, and additional PR recordings in a new monkey (monkey E). The results confirmed our previous findings (results of PR recordings in monkey E are shown in Extended Data Fig. 4; results of muscimol experiments

pooled across two monkeys are shown in Fig. 4 and Extended Data Fig. 21, with results for each individual animal shown in Reviewer Fig. 8).

4) Recordings with a set of perfectly feature-matched unfamiliar faces. Reviewer 2 raised concerns about the extent to which familiar and unfamiliar faces were feature matched. To address these concerns, we generated a new set of unfamiliar faces that were perfectly feature matched to the familiar faces (instead of identifying feature-matched unfamiliar faces post-hoc). We recorded using these new stimuli in Monkey A in TP and Monkey E in both PR and TP. The results strongly confirmed our previous findings (see Monkey E PR in Extended Data Fig. 4d, e).

5) New analysis revealing that axis change, unlike repetition suppression, is robust across experimental contexts. Given our claim that axis change is the mechanism for representing the long-term memory of familiar faces, Reviewer 3 raised an important question about whether axis change is stable across temporal contexts. Previously, we had found that repetition suppression is not (compare Fig. 1c vs. Fig. 3a, Fig. 1g vs. Fig. 3b). We thus repeated our time-resolved analysis of similarity between familiar and unfamiliar axes using data from our first experiment (screening stimulus), and *found that long-latency axis change for familiar faces persisted in this condition* (Extended Data Fig. 19). This shows that axis change, unlike repetition suppression, is robust across different temporal contexts, making it a more likely candidate mechanism for long term memory.

6) Removed claim that neural distance increase is directly related to axis change. We have removed this claim because we now think that the increased neural distance observed in Experiment 2 is mostly due to the increased response magnitude to familiar faces in this experiment. We found that the previous analysis demonstrating increased neural distance due solely to axis change independent of magnitude depended critically on how the axis was computed (spike-triggered average rather than linear regression), and thus is likely not a robust finding. Furthermore, contrary to our observation in face patches AM and PR, TP showed a similar neural distance increase for familiar faces, but did not show any axis change at long latency, further suggesting that increased neural distance is not causally related to axis change.

Referees' comments:

Referee #1 (Remarks to the Author):

This is a tour de force experiment and analysis addressing the fascinating question of how face representation, based on neural axis tuning in a face shape/appearance space, as previously shown by this group, changes between unfamiliar faces (previously studied) and familiar faces (until now not studied). The results clearly show that the tuning axis model fit to a given neuron's responses to 1000 unfamiliar faces explains the early response phase for 36 familiar faces in anterior face patch AM and in a recently discovered perirhinal face patch (PR), but later responses diverge substantially.

We are grateful to the reviewer for his appreciation of our work and valuable suggestions.

The authors characterize this change as a shift in axis direction in the face parameter-based space. But the shifted axis explains a significant amount of variance for only a fraction of neurons (extended data Fig. 9e). The small set of 36 familiar face is not adequate for constraining the 20 parameter model, as the authors recognize explicitly. And the fact that explained variance goes up nonetheless suggest that the new axis model is overfitting the small number of data points.

The reviewer is correct that the explained variance indicates overfitting in Extended Data Fig. 9e of the first version of the manuscript. This arises because the figure showed explained variance for training data (i.e., the faces used to compute the axis). We repeated the same analysis in a cross-validated way, calculating explained variance for faces not used to compute the preferred axis (Reviewer Fig. 1 and updated Extended Data Fig. 11c). The result shows how well the model could generalize to unseen faces, which cannot be explained by overfitting. The proportion of significant cells remains similar to that from the previous analysis.

Reviewer Figure 1. (replicating Extended Data Fig. 11c) Explained variance using a leave-one-out cross validation procedure. Shaded bars indicate the subset of cells for which the explained variance was significantly higher than for stimulus-shuffled data (1000 repeats).

To confirm our main conclusion, we repeated the same analysis as Fig. 2c using only cells identified as significant through the cross-validated analysis above. The result remains similar (Reviewer Fig. 2).

Reviewer Figure 2. Cosine similarity between preferred axes for unfamiliar-unfamiliar (orange) and unfamiliar-familiar (blue), using only cells with significant cross-validated axis tuning (compare to Fig. 2c).

It is clear that a greater axis shift is required to fit/overfit familiar faces compared to 36 unfamiliar faces. But, as discussed below, the alterations in neural responses are potentially explainable with something simpler than the 20 coefficient changes used here. The authors conclude that familiar faces induce a delayed shift in tuning axis direction in AM and PR, while at the same time admitting that they cannot rule out a different explanation in terms of nonlinear tuning changes. That is a disappointingly tentative conclusion.

But I think the authors could extract much more definite results from this magnificent dataset. The first alternative to explore is nonlinearity, which the authors mention frequently but believe cannot be tested here. (They also seem to say that nonlinearity could only explain the residual variance after axis fitting, but axis overfitting could capture nonlinearities.) But a general form of nonlinearity could be implemented here by just passing the dot product from the original 1000 unfamiliar face-based model through a final, fitted logistic function. If nonlinearity emerges, then the shape of the logistic function should change across time, from basically linear to more of a step function. Delayed nonlinearity is a simple change, describable with simple models, and already known to occur in IT on a delayed timeframe as observed here, following initially linear responses, also as observed here (Brincat & Connor, Neuron 2006). This would be mechanistically far simpler than changing every neuron's tuning direction in the face space. And, nonlinear interactions between features/parts are known to increase with stimulus exposure in IT (Baker, Behrmann, & Olson, Nature Neuroscience 2002). The sparsely responding neurons mentioned in the text might also be captured by a face space axis combined with a nonlinear logistic function; they might actually represent one end of a continuum of nonlinearity in the AM and PR populations. The logistic function cannot capture the full complexity of potential nonlinearities in the 20 dimensional space, but it can capture the overall supra- or sub-additivity they may produce.

The reviewer suggests a "general form nonlinearity" model: a logistic nonlinearity after linear projection. To test this, we performed the analysis suggested by the reviewer. We computed explained variance for responses to 36 familiar or 36 unfamiliar faces by the preferred axis computed from 964 (= 1000 - 36) left-out unfamiliar faces with either a linear function or a logistic function ($f(x) = \frac{L}{1+e^{-k(x-x_0)}}$) (Reviewer Fig. 3). The result shows that the extra explained variance captured by the logistic function is very small (middle row), compared to that caused by axis change (bottom row).

Reviewer Figure 3. (replicating Extended Data Fig. 17a) Testing whether a logistic output nonlinearity can explain apparent axis change. We compared the explained variance of an axis model followed by a linear function or logistic function. 36 unfamiliar or 36 familiar faces were first projected to a preferred axis computed from responses to 964 unfamiliar faces, then the projected value vs. neural responses were fitted by either a linear function or a logistic function ($f(x) = \frac{L}{1+e^{-k(x-x_0)}}$). Top: explained variance for 36 unfamiliar (y-axis) or 36 familiar (x-axis) faces using axis model with linear output function (each dot is one cell, $N = 134$ cells for AM and $N = 72$ cells for PR). Middle: explained variance for 36 familiar faces using axis model with linear output function (y-axis) or a logistic output nonlinearity (x-axis); the latter values are only slightly higher. Bottom: explained variance for 36 unfamiliar faces using axis model with linear output function (y-axis) or 36 familiar faces using axis model with a logistic output nonlinearity (x-axis); The slight increase in explained variance obtained by applying a logistic output nonlinearity cannot undo the decrease caused by axis change (however, explained variance is similar using familiar axes for familiar responses and unfamiliar axes for unfamiliar responses, Extended Data Fig. 11c).

The reviewer suggests that a delayed nonlinearity “would be mechanistically far simpler than changing every neuron’s tuning.” We wish to clarify that tuning change is a functional description, not a claim about

underlying mechanism. The observed axis change could result from local recurrent connections (e.g., as proposed in Brincat & Connor, Neuron 2006). We now clarify:

“We speculate that by lifting representations of face memories into a separate subspace from that used to represent unfamiliar faces (Fig. 3g), attractor-like dynamics may be built around these memories through a recurrent network to allow reconstruction of familiar face features from noisy cues without interfering with veridical representation of sensory inputs (Brincat & Connor, Neuron 2006; Hopfield, PNAS 1982).”

In summary, we find that a simple output nonlinearity cannot explain our results. More general forms of nonlinearity would be more complicated than a linear change-of-axis model, since the number of nonlinear terms increases exponentially with dimensionality (e.g., for a 20 parameter model, including second order interactions would increase the number of parameters to 400).

Another hypothesis much simpler than changing tuning in 20D space is that neurons become less sensitive to features that are not diagnostic for the small group of individuals they know. This could be modeled easily by deleting axes in the original tuning space to search for a lower dimensionality in which a given neuron operates to discriminate just 36 individuals. This would avoid fitting new values, and it would again have a known type of neural implementation, similar (or perhaps identical) to attentional suppression of distractor information. It would also explain the lower average responsiveness to familiar faces under at least some conditions.

To check whether tuning to familiar faces can be explained by loss of tuning to certain dimensions, we computed the sensitivity of each feature for 36 familiar faces or a random subsample of 36 unfamiliar faces. Sensitivity was measured as the slope of the linear regression of feature value vs. neuronal firing rate. We then plotted the 20 feature sensitivities for familiar vs. unfamiliar faces (Reviewer Fig. 4, top panel). The dots in the blue rectangles (corralling points for which sensitivity to the familiar feature goes to ~ 0) agree with the Reviewer’s hypothesis, while the dots in the red rectangles indicate that cells became *more sensitive* to some dimensions of familiar faces. On average, sensitivity for familiar faces was larger than that for the unfamiliar faces (Reviewer Fig. 4, bottom panel).

Reviewer Figure 4. (replicating Extended Data Fig. 17b) Comparing sensitivity to features in familiar vs. unfamiliar faces. Top: scatter plot of 20 feature sensitivities from 134 AM cells and 72 PR cells, for familiar (y-axis) and unfamiliar (x-axis) faces. Bottom: Distribution of feature sensitivity values for familiar and unfamiliar faces.

We now include the results shown in Reviewer Figs. 3 and 4 in the revised manuscript:

“Could the apparent axis change be explained by a simpler change, e.g., an output nonlinearity or sensitivity change, without necessitating a change in axis? To address this, we fit the data to these simpler models, and confirmed that they could not explain cells’ responses to familiar faces as well as axis change (Extended Data Fig. 17).”

Even if changing direction in the space is essential, this could also be constrained by using some metric to minimize the number of coefficients that are changed. Just as distractor responses can be suppressed by attention, target responses can be magnified. Each dimension/coefficient in these analyses corresponds to some combined set of measurable changes in face appearance. Some of those differences might be more diagnostic for the set of 36 familiar individuals. Changes in just a few coefficients could be adequately constrained by 36 data points, avoiding overfitting, and thus potentially achieving more significance in the randomization test.

We thank the reviewer for this great suggestion. As suggested by the reviewer, to reduce the number of coefficients used to compute familiar axes, we hypothesized that the familiar axis is a linear combination of the unfamiliar axis and a few additional axes that are diagnostic for the set of 36 familiar faces. We used principal component analysis for the 36 familiar faces in the 20d feature space to extract the most diagnostic axes from the familiar faces (P_i , i th principal component of familiar features). Then for each cell, if the first n familiar PCs are used, the number of coefficients to compute is $n+1$ (n coefficients for the n familiar PCs plus one coefficient for the unfamiliar axis).

$$r_{familiar} = \sum_{i=1}^n a_i P_i + a_{n+1} \tilde{c}_{unfamiliar}$$

Finally, to evaluate the resulting axis, we computed the explained variance of each cell using the axis. As shown in Reviewer Fig. 5 (top), the population average explained variance initially increases ($n = 1$ to 5), but then plateaus. Thus indeed, as the reviewer suggests, we need only a small number of dimensions to explain the neural responses on average. However, we did not see overfitting when we increased the dimension to 20 (the explained variance plateaus instead of decreasing), and the number of significant cells was negligibly decreased for AM (20d, 50/134; 5d, 49/134) and showed no change for PR (20d, 31/72; 5d, 31/72). Also, examining the single cells, we do see many cells significantly tuned to diagnostic dimensions beyond the top five (Reviewer Fig. 5, bottom). To simplify the story, we did not include this analysis in the revision.

Reviewer Figure 5. Cross validated explained variance of familiar axes in a low dimensional subspace. Top: Population average of positive explained variance across cells vs. number of familiar PCs used to expand the subspace for familiar axis, error bar, SEM. Bottom: Example cells for the same analysis.

These analyses, alone or in combination, could also produce results that are clearly interpretable. For a given neuron, the few dimensions that best capture the data could be cashed out as visualized changes in face shape/appearance. Those dimensions, across neurons, could tell us about what specific feature combinations are used to identify familiar faces, and whether those specific features are particularly diagnostic for the individual faces in the familiar group. That would add a lot to the scientific interest of this paper. At the least, such analyses could rule out alternatives to the authors' favored hypothesis of tuning direction changes in 20D space.

We again thank the reviewer for the valuable ideas. We fully agree that these new analyses give us a deeper understanding of the results.

Referee #2 (Remarks to the Author):

She et al. conducted a study to investigate how responses in macaque AM face patch and perirhinal cortex differ for familiar and unfamiliar faces. The main finding is that familiar faces were differently represented than unfamiliar faces at different latencies. The results presented in this study will be of interest to researchers studying face and object recognition. What impresses me about the manuscript is the thoroughness with which the research was done and the presentation of both raw data and higher-level summaries.

We are grateful to the reviewer for their appreciation of our work and valuable suggestions.

However, I see several issues with this manuscript that I would invite the authors to address.

- The main finding is that familiar and unfamiliar faces are represented at different times in two brain regions. However, there is a potential flaw in this interpretation. In figure 2e there seems to be a difference in contrast between familiar and nonfamiliar faces. Unfamiliar faces seem paler than familiar faces (extended data figure 2) for both human and monkey faces. The authors say that they compared the similarity of features in their feature model, however, this is not a standard model used to evaluate features. I would like to invite the authors to check how features of familiar and unfamiliar images differ in some of the standard low-level feature models like GIST, HMAX, or early layers of a deep convolutional neural network (be it AlexNet or other architecture).

Thank you for raising this important point. We computed feature distributions for familiar and unfamiliar faces using GIST, HMAX, and AlexNet layer 1. The distribution of feature values was highly overlapping for familiar and unfamiliar faces (Reviewer Fig. 6a). However, we did see greater distances for familiar faces in the low-level feature space. To resolve this issue, we performed a new analysis on the previous data as well as a new experiment.

New analysis: We checked to see if there is a correlation between face distances in the low-level feature space defined by AlexNet layer 1 and face distances in the AM neural space. We found no correlation for familiar faces and very weak correlation for unfamiliar faces (Reviewer Fig. 6b). This shows that low-level feature differences cannot account for our observation of increased neural distances between familiar faces.

New experiment: Second, in order to completely remove all differences in low-level features between familiar and unfamiliar faces, since the original submission, we performed new recordings in PR and TP

using new unfamiliar faces that were matched in both low-level and high-level features (Reviewer Fig. 6c; also see point #4 in “General summary of the revision” at the beginning). This new stimulus set is described in the Methods, Stimulus Set #5. The results in PR remained consistent with what we previously reported (Monkey E in Extended Data Fig. 4d, e).

Finally, regarding the Reviewer’s specific concern of a contrast difference between familiar and unfamiliar faces, please note that in Fig. 2b (inset), we specifically measured the axis for faces at two different contrast levels and found that this produced no change in axis.

Reviewer Figure 6. Controlling for low-level feature differences between familiar and unfamiliar faces. (a) Distributions of low-level feature values for familiar (blue) and unfamiliar (orange) faces, computed using GIST, HMAX layer C2, and AlexNet Layer 1. (b) Scatter plot of AM neural distances versus AlexNet layer 1 distances for all pairs of familiar (left) and unfamiliar (right) faces. (c) Distributions of AlexNet layer 1 distances of all pairs of 36 familiar faces and newly generated 28 sets of 36 unfamiliar faces, confirming feature matching.

- When the authors mention that they defined 30 familiar and unfamiliar faces that were feature-matched, they should describe in the results section how this feature match was determined (apart from discussing it in details in the methods section).

Thank you for this suggestion. We now describe in brief in the results section how the feature match was determined:

“As a second control, to ensure that the effects were not due to differences in the feature content of familiar versus unfamiliar faces, we identified 30 familiar and 30 unfamiliar faces that were precisely feature-matched. In brief, we used gradient descent to search a subset of familiar and unfamiliar faces that were matched in the variance and distribution of each feature, and in the distribution of pairwise face distances (see Methods and Extended Data Fig. 10).”

- There is a different number of familiar and unfamiliar faces (34/16 and 36/100 in two experiments). While it is understandable that there is a limitation in the number of familiar faces that the animals were exposed to, I was wondering why the number of unfamiliar faces was not matched to that of familiar faces. It would be helpful if the authors could address that.

In Experiment 1, our goal was to compare response amplitudes to familiar and unfamiliar faces in AM and PR. Originally, we wanted to test not only personally familiar but also pictorially familiar faces (in case this affected responses). Thus we matched the number of faces *within each familiarity category*: 9 personally familiar human faces, 8 unfamiliar human faces, 9 personally familiar monkey faces, 8 unfamiliar monkey faces, plus 8 pictorially familiar human faces and 8 pictorially familiar monkey faces. Since the results with pictorially familiar faces were largely the same as those with personally familiar faces, we decided not to show them in order to reduce paper complexity.

In Experiment 2, our goal was to measure axis tuning for familiar and unfamiliar faces. In order to get a quality estimate of the preferred axis for unfamiliar faces, we presented 1000 unfamiliar faces (as in Chang and Tsao 2017). Importantly, in Fig. 2b, c, we used a matched number of left-out unfamiliar faces and familiar faces (36).

- It is hard to follow the results section where the authors discuss the axis code without diving into the methods. There should be more explanation of what axis code is in the results section so that the reader can interpret the data.

We have now added a more detailed description of the axis code to the results section of the revised manuscript:

“According to the axis code, face cells in IT compute a linear projection of incoming faces formatted in shape and appearance coordinates onto specific preferred axes³; for each cell, the preferred axis is given by the coefficients \vec{c} in the equation $r = \vec{c} \cdot \vec{f} + c_0$, where r is the response of the cell, \vec{f} is a vector of shape and appearance features, and c_0 is a constant offset (see Methods). Shape features capture variations in the location of key facial landmarks (e.g., outline, eye, nose, and mouth positions, etc.), while appearance features capture the shape-independent texture map of a face³.”

- In the discussion section, it is mentioned that a simple computational model could recapitulate the study’s findings but this finding is not described in the results section, so please describe it if this result is to be discussed.

In order to reduce the complexity of the manuscript, we have decided to remove this computational model.

- In the discussion section, the authors state that their results contradict some of the current literature but they do not specify where these differences may have come from. In general, the discussion section would benefit from more details on the relevant literature and its relation to the current study.

We think the main reason previous studies did not observe axis change is that they did not know to look for it. The mechanism for encoding face identity through projection onto shape and appearance axes (Chang and Tsao, 2017) was previously unknown, so researchers did not have the conceptual framework to compare preferred axes for familiar and unfamiliar faces/objects—and thus simply compared

differences in firing rates; furthermore, when comparing firing rates, previous researchers did not have the technical capability to record from the same cells before and after familiarization, and thus assumed that rank order preference was preserved (e.g., Woloszyn and Sheinberg 2012). In a nutshell, a change in mean firing rate to familiar faces—either up or down—was the only thing that previous studies were equipped to find, given limitations in the conceptual framework for IT coding as well as technical capabilities. In addition, as we point out in the introduction, unlike our study, previous recordings exploring neural correlates of visual familiarity were not targeted to specific subregions of IT cortex known to play a causal role in discrimination of the visual object class being studied. This could also have contributed to the differences. We now discuss the relation of our study to previous studies more extensively in the revised manuscript:

“Previous physiological work on representation of familiar stimuli has focused largely on repetition suppression, the observation that the response to familiar stimuli is reduced⁵⁻¹⁰. We found familiarity could be detected even before emergence of repetition suppression, enabled by a very short latency shift in the overall response subspace. In contrast, repetition suppression was not a robust indicator of familiarity in any of the three patches. Instead, relative response amplitude to familiar versus unfamiliar faces was highly sensitive to temporal context (cf. Ref. ⁴¹ showing both repetition suppression and enhancement in IT cortex during working memory). In a temporal context where familiar face responses were enhanced, we found improved decoding accuracy for familiar faces⁴², consistent with face recognition behavior.”

- The study would benefit from more discussion about the effects of familiarity and the time course of representations. For example, Dobs et al. 2019 has studied the effects of familiarity in the MEG responses in humans but this study is not cited.

Thank you for this suggestion, we now cite this study:

“In a temporal context where familiar face responses were enhanced, we found improved decoding accuracy for familiar faces (Dobs et al., 2019), consistent with face recognition behavior.”

Minor points:

- The figures have too many subpanels (e.g., labelled e1, e2, e3, e4). I would label them with successive letters rather than a mix of letters and numbers.

We have renamed all subpanels with successive letters in our revision.

- In extended data figure 1, the colour bar should be labelled in the figure. It is not clear why the colorbar is not within the same range for all monkeys.

The color bar is now labeled ($-\log_{10} p$, where p is the significance of a t-test comparing activation to faces versus non-face objects) in the revised manuscript. Due to differing ages, different monkeys have different sizes of temporalis muscle, which greatly affects fMRI signal quality. For older animals, we need to use a lower threshold to see face patches. In addition, signal-noise-ratio is affected by amount of MION contrast agent injected, distance of coil to the monkey's brain, etc. Importantly, all face patches were confirmed by single-unit recording.

- It would be easier for reviewers if the authors embed figures in the text for the potential revision.

We have uploaded a version of the manuscript with all figures embedded in the main text as a supplemental file.

Referee #3 (Remarks to the Author):

Many studies have shown that neurons in the inferotemporal (IT) cortex represent the sensory percept of visual objects, more specifically, neurons clustered in face patches within IT have been shown to

represent visual faces. But do these same neural populations represent the memory of visual objects or faces? In this manuscript the authors tackle this question by focusing on long-term memory, and recording from three face patches, two in IT (face patch AM and ML) and one face patch in perirhinal cortex (PR). To investigate how long-term memory might be represented in these face patches they compare neural responses to images of familiar and unfamiliar objects. Personally familiar images depicted people, monkeys, and objects that the animals interacted with on a daily basis. To confirm familiarity, two of the five monkeys studied were tested behaviorally, and showed highly significant preferential looking toward unfamiliar face stimuli and away from familiar stimuli. It is important to note that no animals were behaviorally tested to see if they are better at identifying familiar faces from each other.

This paper is not lacking in results. Many of the results are interesting, although I am not sure how much interest these results will garner outside of the community that studies visual object/face recognition. In truth it would take a serious study of previous work before many of the results can even be understood. I commend the authors for attempting to fit all their results in a short manuscript. Unfortunately, this doesn't come without a heavy toll on the reader. Understanding each result requires tracking data analyses and ideas interspersed between text, sub-figures, supplemental text, and supplemental sub-figures. In tracking these results, I was left with more questions than answers and I am not convinced that they cracked the neural code for face memory. I do believe that they propose a different way of looking at encoding face memory, and given the room in a longer manuscript in a more specialized journal they might be able to present a more cohesive story that systematically goes through their different manipulations and results. A longer manuscript would allow them to more clearly explain their challenges to previous ideas and results on how familiarity impacts the neural code for objects and faces in IT.

We are grateful to the reviewer for their appreciation of our work and valuable suggestions. We have significantly streamlined the presentation in the revised manuscript. We underscore that *our central story is a simple one*, namely, that face memory is encoded through a long-latency change in axis. This challenges a prevailing assumption about how long-term memory is represented, the repetition suppression hypothesis. Given the fundamental importance of long-term memory to animal behavior, and our almost complete lack of knowledge about how it is actually represented in the brain, we believe the discovery of long-latency axis change for familiar faces is an important advance. Indeed, one of the most common questions the senior author is asked when giving talks about IT cortex is “Do IT cells care only about visual shape, or do they also care about the meaning?” Meaning arises from associative relations within a large web of knowledge, and familiarity constitutes one of the most powerful forms of meaning—relating an item to our own previous experience. Our findings show for the first time that both shape and meaning can be represented by the same IT cell population via temporal multiplexing. Thus we believe the findings should be of interest to a broad community beyond experts in face/object recognition.

Based on the title of the paper and the abstract, the authors argue that the novelty of their study lies in their claim that it uncovers “The neural code for face memory”. In this neural code familiar faces are represented in distinct neural subspace that at short latency is shifted relative to the subspace for unfamiliar faces and then distorted in order to increase neural distances between familiar faces at long latency. In the absence of behavioral data that would confirm that monkeys are better at identifying familiar faces from each other, a more appropriate title would be “A potential neural code for long-term face memory”.

We thank the reviewer for this important suggestion. We have trained two monkeys to perform a face

identification task and found that the monkeys were significantly better overall at identifying familiar faces (see revised manuscript, Fig. 4d and Methods). Please note that we have removed the claim regarding the relationship between axis change and increased neural distance (please see point #6 in “General summary of the revision” at the beginning).

The proposed neural code is within the framework of the “axis code”, which is an elegant geometric approach to the neural coding of face identity described in an important previous publication by the same lab. Using this framework, the authors show that familiarity causes neural responses in AM and PR (even ML based on my reading of extended figure 20) to go off axis. The authors assert that this axis change serves to increase the neural distance between familiar faces.

The crux of the authors claim for a link between the axis change and the increase in neural distance is based on an analysis shown in figure 2h2 in which they show the mean pairwise neural distance between 30 familiar faces is larger when using a familiar axis compared to using an unfamiliar axis. The result is very clear in figure 2h2, although I would have liked to see the same analysis done for the unfamiliar faces for comparison.

We have collected neural data from one more face patch TP during the manuscript revision, and performed more comprehensive analyses including those suggested by the reviewers. Considering all the evidence we have now, we conclude that there is no causal relationship between axis change and increase in neural distance (see point #6 in “General summary of the revision” at the beginning). Thus we have revised the manuscript to remove the claim. We wish to underscore that the central discovery of long-latency axis change is important because it shows that *the brain encodes memories using the same cells, but a different code, than that for representing visual features*. Moreover, the new negative result in TP emphasizes the non-triviality of this discovery, showing that long-latency axis change occurs only in a circumscribed set of temporal lobe regions.

I would have also preferred that the authors didn’t jump from looking at 9 familiar faces in one analysis, to 36 familiar faces in another, to 30 completely different faces in yet another. All of the results are presented in one figure and assumes without verification that if the same analysis was applied to a different image set it would produce the same results. What makes this even more complicated is that some of the data were collected after the monkeys spent a month learning/watching videos of 27 faces to make them familiar.

We have made the presentation of Fig. 2 more streamlined in the revision. Our original reasoning was as follows: (i) we began with 9 familiar faces in Fig 2a for the purpose of narrative continuity from Fig. 1 (where we hadn’t yet introduced the cinematically-familiar faces used for Experiment 2); importantly, the result shown in Fig. 2a still holds if using all 36 familiar faces (or subset of 30 feature-matched faces); (ii) then, to compute the preferred axis, we wanted to use all available familiar faces to maximize the use of all data for Fig. 2b, d, e (revised manuscript Fig. 2 b-d); importantly, the results hold true for the subset of 30 feature matched faces; (iii) finally, we used the 30 feature matched faces for Fig. 2f-h (revised manuscript Fig. 4a-c), because these analyses measure face distances and decoding performance, and would be meaningless if the familiar and unfamiliar faces were not feature matched (since then differences could simply have arisen due to feature differences rather than familiarity). We hope the Reviewer finds the new figure organization clearer and easier to understand.

Importantly, 1) in Extended Data Fig. 11b, d, we show that the preferred axis computed using split-halves is consistent, and 2) our new recordings in PR and TP used an *entirely new set of unfamiliar faces* that

were designed to be perfectly featured-matched to the familiar faces, and all of the results remained robust when using this new set of unfamiliar faces. This explicitly confirms that *the same analysis applied to a different image set produces the same results* (see point #4 in “General summary of the revision” at the beginning).

There are two potential issues with the link between the change in axis and the increase in neural distance between familiar faces:

1-Neurons in areas AM and PR show a change in axis with a commensurate change in face pairwise neural distance. But the data is pooled across animals, 3 animals in the case of AM and 2 animals in the case of PR. I would have liked to see this analysis broken down by monkey especially since monkey B doesn't show a big change in axis.

We added plots of neural distance for individual monkeys to Extended Data Fig. 4 in the revision as requested. We also recorded from PR in one additional monkey (monkey E), which showed results consistent with our previous findings, specifically showing a large change in axis at long latency (Extended Data Fig. 4d, e).

2-Neurons in area ML(extended figure 20d), at least to my eye, shows a change in axis (blue histogram for familiar faces is shifted from the red histogram (unfamiliar faces)), without a commensurate change in neural distance between familiar faces. ML is based on data from one animal. The authors claim that “Overall, representations of familiar versus unfamiliar faces were much more similar in ML compared to AM (Extended Data Fig. 20d-k)”, so I might be missing something. From a graphical standpoint the changes in cosine similarity would be much easier to evaluate and compare if they were plotted as a scattergram in order to see the axis change on a neuron by neuron basis. The histogram hides whether the axis change applies to all cells or is mostly driven by a few cells that show large changes.

First, we want to clarify that the ML data is from two animals, monkeys A and D. There is no corresponding MRI image in Extended Data Fig. 1 showing the targeted recording in ML from monkey A because the recording was performed using an early version of Neuropixels NHP probe which is not MRI compatible; we added a note about this to the figure legend for Extended Data Fig. 1 in our revision.

The reviewer makes an important and correct observation that there is significant axis change in ML without a commensurate change in neural distance between familiar faces. This suggests that axis change can serve other purposes besides increasing neural distance between faces for the purpose of face recognition. For this and other reasons already explained above, we have removed the claim of a causal relationship between axis change and increased neural distance.

Finally, following the reviewer's suggestion, we computed scatterplots of cosine similarity for familiar versus unfamiliar faces (Reviewer Fig. 7); the plots make clear that the decrease in cosine similarity for familiar faces is not driven by only a few cells.

Reviewer Figure 7. Scatter plot of cosine similarities for unfamiliar-unfamiliar (y-axis) versus unfamiliar-familiar (x-axis) axes (same data as Fig. 2b); each dot represents one cell.

We greatly appreciate and agree with the Reviewer’s observant comment re axis change in ML, and have now substantially modified our discussion of the ML findings:

“Do signatures of familiarity coding, as observed in AM, PR, and TP, exist even earlier in the face patch pathway? We mapped responses to familiar and unfamiliar faces in face patch ML, a hierarchically earlier patch in the macaque face processing pathway that provides direct input to AM^{21,28}. Responses to the screening stimuli in ML exhibited a similar pattern as in AM, showing suppression to personally familiar faces at long latency (Extended Data Fig. 22a, c). However, population representation similarity matrices did not show distinct population responses to familiar versus unfamiliar faces (Extended Data Fig. 22b). Furthermore, the population average firing rate showed a sustained divergence between responses to familiar and unfamiliar faces much later than in AM (160 ms compared to 140 ms in AM, Extended Data Fig. 22c), suggesting that ML may receive a familiarity-specific feedback signal from AM. **Importantly, ML neurons also showed axis divergence (Extended Data Fig. 22d-f), consistent with the idea that memory is stored across the same hierarchical network used for representation**³⁹. Finally, familiarity could be decoded in ML even earlier than in AM (Extended Data Fig. 22g-k), and ML also showed a boost in identity decoding performance for familiar faces, albeit smaller than that in AM, PR, and TP (Extended Data Fig. 22i, j). **Overall, these results suggest that ML also plays a significant role in storing memories of faces.**”

Even if the change of axis does indeed increase the neural distance between familiar faces, in the absence of behavioral data that tests face identification performance for familiar and unfamiliar faces and shows that the difference in identification performance is predicted by the neural effects reported, it is premature to claim that the proposed code is “the neural code for face memory.”

As already addressed above, we performed a new behavioral experiment in two animals to quantify face identification performance for familiar versus unfamiliar faces, and both animals showed better recognition of familiar faces at certain difficulty levels (revised manuscript Fig. 4d and Methods). We underscore again that we believe the long latency axis change is exciting because it is a completely unexpected and strong effect that represents a new mechanism for encoding visual memory beyond repetition suppression, and can co-exist with feedforward feature representation due to temporal multiplexing.

The authors also show that inactivation of PR using muscimol doesn’t impact the change in axis-code

between familiar and unfamiliar faces in AM. Inactivation of PR does change the neural responses in AM. It isn't clear whether inactivation of PR impacts the neural distance between familiar faces, or whether it changes the similarity matrix. As far as I can tell, these data were also only collected in one animal.

Indeed, we only performed the muscimol experiments in one animal for the initial manuscript. We now collected data from a second animal (monkey A), and the results were consistent with those in the first animal (monkey E) (Reviewer Fig. 8).

Specifically, inactivation of PR did not have a salient effect on either neural distances between familiar and unfamiliar faces (Reviewer Fig. 8a) or on the similarity matrix (Reviewer Fig. 8b).

Reviewer Figure 8. (a) Pairwise face distances for individual subjects (orange, unfamiliar; blue, familiar) and (b) similarity matrix before and after inactivation of PR with muscimol. The similarity matrix was computed using responses before and after inactivation that were z-scored separately.

I agree with the authors that their results pose a challenge to repetition suppression as a model for familiarity encoding. It is quite interesting that repetition suppression depended heavily on temporal context. In the screening stimulus set where familiar monkey faces and human faces and objects were mixed with their unfamiliar counterparts, all familiar object categories show a delayed suppression. However when familiar monkey faces were presented with a 1000 unfamiliar faces, the familiar faces showed enhanced responses. In the context of short term memory, it has been shown that not only does repetition suppression depend on temporal context, but it is also heavily modulated by task (Miller and Desimone, 1991,1994).

We thank the reviewer for noting the Miller and Desimone work, which indeed is relevant in showing that repetition suppression can be increased by a short-term working memory task beyond what is expected from temporal contiguity (Miller and Desimone 1991), and furthermore, that some cells can show memory-dependent repetition enhancement depending on task (Miller and Desimone 1994). We note that, to our knowledge, no one has ever reported that the *same cells* can show either repetition

suppression or enhancement depending simply on stimulus presentation statistics. We now cite the Miller and Desimone 1994 study in the Discussion:

“Previous physiological work on representation of familiar stimuli has focused largely on repetition suppression, the observation that the response to familiar stimuli is reduced^{5–10}. We found familiarity could be detected even before emergence of repetition suppression, enabled by a very short latency shift in the overall response subspace. In contrast, repetition suppression was not a robust indicator of familiarity in any of the three patches. Instead, relative response amplitude to familiar versus unfamiliar faces was highly sensitive to temporal context (cf. Ref. ⁴¹ **showing both repetition suppression and enhancement in IT cortex during working memory**).”

But is the axis-code shift and distortion immune to such response modulation?

We thank the reviewer for raising this very important question. To address it, we computed the rank order correlation between neural responses to the set of same faces (9 personally familiar and 8 unfamiliar) that were presented in both Experiments 1 and 2 and used the rank order correlation between half trials of the same experiments as a control (Reviewer Fig. 9a). We found that the tuning correlations across experiments were comparable to the trial-trial variations within the same experiment, indicating that context modulation only affects the overall firing rate (scaling), but not the tuning. Thus the axis remains the same across context. To further confirm this, we performed the same analysis as in Fig. 2c for the small set of faces in Experiment 1, but in a smaller face space (5-d), and found that a similar effect as in Fig. 2c can be seen (Reviewer Fig. 9b).

Reviewer Figure 9. Testing robustness of familiar and unfamiliar axes across experimental contexts. (a) Left: Correlation in rank order of familiar face stimuli between split halves of trials (y-axis) is plotted against correlation between rank order of the same faces between Experiments 1 and 2; each dot represents one cell. Right: same for unfamiliar faces. (b) Axis change for small set of faces from Experiment 1 (9 personally familiar faces and 8 unfamiliar faces) vs. large set of unfamiliar faces from Experiment 2 using the subset of cells that were recorded using both stimulus sets; same analysis as in Fig 2c, using a 5-d face space.

The authors attempt to address this by measuring responses to low-contrast faces. They show that axis-similarity was not changed by the contrast-varied faces which were meant to decrease response gain.

However, the authors neglect to share with us how big the change of contrast was and how it actually impacted the responses of the neurons.

The low contrast stimuli were at 50% of the contrast of the original stimuli (Reviewer Fig. 10a), and the change in contrast had a marked effect on firing rates (Reviewer Fig. 10b, c) though not on preferred axis (Fig. 2c).

Reviewer Figure 10. Effect of contrast on face cell responses. (a) Example face stimuli at full and 50% contrast. (b) Mean responses of two example AM cells to 1000 full and 50% contrast face stimuli. (c) Average response across 24 cells from face patch AM.

I was also struck by the analysis the authors performed in Extended figure 18. The authors focus on the fact that the difference between responses to 9 personally familiar and 8 unfamiliar faces depends on the larger stimulus set used. But what caught my attention was that the enhancement they describe in Fig 3b in their “thousand faces” experiment, seems to be largely driven by the cinematically familiar faces. These faces are the newly “learned” familiar faces. The personally familiar faces show less enhancement if any. When plotting the axis tuning example cells in Fig2a1, I am puzzled by why they restrict themselves to the 9 personally familiar, and why they don’t include all familiar faces. I also wonder how much the changes in preferred axis is actually driven by the enhancement in responses to the cinematically familiar faces?

The reviewer is correct that response enhancement in Experiment 2 occurred predominantly for cinematically familiar faces. Importantly, however, the axis change was still present when we analyzed non-cinematically familiar faces (Reviewer Fig. 11). This, together with the above analysis showing robustness of axis change across context (Reviewer Fig. 9), demonstrates that axis change cannot be driven purely by enhancement of responses.

Reviewer Figure 11. (replicating Extended Data Fig. 18c) Axis change for 17 non-cinematically familiar faces (9 personally familiar and 8 pictorially familiar faces); same analysis as in Fig 2c, using a 10-d face space.

We have now added this analysis to Extended Data Fig. 18c.

The authors address this issue by conceding that their 36 unfamiliar faces are not necessarily white (uncorrelated in feature space). They try a “whitening control” (Extended figure 15b) which didn’t affect their results, but as they point out in the discussion, focusing on 36 familiar faces limits their ability to fully characterize the distortion in the linear subspace that they claim is produced by familiarity.

To clarify, the whitening control was not performed to address whether axis change is driven by response enhancement, but rather to ensure that axis change was not an artifact of feature correlations. In the discussion we pointed out that 36 familiar faces limit our ability to fully characterize the distortion if it is nonlinear; however, we do think 36 faces are good enough to characterize the linear distortion or linear approximation of a nonlinear distortion.

Is the axis code itself immune to stimulus presentation time and stimulus design? As the authors note in their supplemental text, Koyano et al. (2021) using different stimuli and different presentation times show that feature tuning in face cells can be V-shaped. The authors speculate that the V-shape tuning observed is due to an over-representation of the average face in the stimulus set because they continue to see ramped tuning even when they increase their stimulus duration. But the dependence of ramp tuning on stimulus set/ temporal context, makes it difficult to argue that it is more robust than repetition suppression.

Our findings indicate that ramp-shaped axis tuning is robust across both stimulus set and temporal context. Specifically, (i) our new analysis of correlation in preferences for familiar and unfamiliar faces across Experiments 1 and 2 (Reviewer Fig. 9) demonstrates that ramp-shaped tuning is robust across temporal context; and (ii) Extended Data Fig. 11b shows that axis tuning is robust to exact choice of face stimuli.

We believe Koyano et al. were likely confounding feature tuning (which is robust) with repetition suppression. As an analogy, one can imagine testing orientation tuning in a V1 cell with a set of different orientations, one of which is repeated much more often than the others. One would then see adaptation to the repeated stimulus, resulting in a dip in the standard Gaussian, and conclude that orientation tuning is stimulus and temporal context dependent.

In setting up their results the authors pitch two extreme schemes for representing familiar faces (fig 1a), in the first faces are represented as points on a continuous low-dimensional object space and the other is discontinuous, “sparsened”, with separate attractors representing the different objects. They argue that their results challenge the “sparsification” model. But considering that they found a small population of sparse cells in both AM and PR that respond most strongly to images of particular familiar individuals, it is very hard for me to take the challenge seriously. Furthermore, I would be interested to look at analysis of sparseness for the screening set to see if there is difference in sparseness when comparing responses to familiar versus unfamiliar objects and faces. To my eye, the responses similarity matrices in Fig 1d shows hints of changes in sparseness and it would be informative to quantify those changes to see if they are consistent with previously reported sparseness changes with familiarity (ref 15-20).

Following the reviewer’s suggestion, we computed distributions of sparseness from responses to the 18 personally familiar faces and 16 unfamiliar faces from the screening set (Reviewer Fig. 13). The distributions did not show any significant difference (Kolmogorov–Smirnov test, AM $p = 0.74$, PR $p = 0.45$).

Reviewer Figure 13. Distributions of Gini coefficients (a measure of response sparseness) computed for responses to familiar and unfamiliar faces from the screening set.

Moreover, on the topic sparseness, the authors argue that changes in sparseness (“sparsening”) can’t explain why responses to familiar faces go off-axis. Unfortunately, I found Extended Figure 8 to be a bit opaque in making that case. A change in sparseness doesn’t preclude a change in rank order. If the changes in sparseness associated with increases in familiarity include a change in which image the cells respond best, such a change would be consistent with the off-axis response described in this paper. In

general I found the schematic figures (Figure 1a, Figure 3g and Extended Figure 8) to be detached and unrelated to the actual data analysis figures used to support the same ideas.

We apologize for causing confusion here. We meant only to say that *monotonic* sparsening cannot explain our results. Previous theoretical models of sparsening driven by familiarity have, to our knowledge, assumed monotonic sparsening, in which the rank order of preferred stimuli is preserved (e.g., Lim & Brunel Nature Neuroscience 2015 model sparsening as a monotonic transform arising from an increased threshold for low-firing inputs and a decreased threshold for high-firing inputs). We now clarify in the revised manuscript that only monotonic sparsening models would be inconsistent with our findings:

“This departure in AM and PR was not a simple gain change: the strongest responses to familiar faces were often to faces projecting somewhere in the middle of the ramp rather than on the end (Fig. 2a). Thus this departure cannot be explained by an attentional increase or decrease to familiar faces, which would elicit a gain change³⁴. Indeed, the effect cannot be explained by any monotonic transform in response, such as repetition suppression or **monotonic sparsening**^{8,10}, as any such transform should preserve the rank ordering of preferred stimuli (Extended Data Fig. 9b).”

I understand the challenge of working with non-human primates. Especially considering the numerous results presented in this manuscript. I found it extremely difficult to figure out which monkey contributed to what results.

We have included main analysis for each subject in Extended Data Figure 4 except muscimol experiments.

Many of the conclusions are weakened by the fact that results represent data from only one monkey. As far as I can tell, the AM muscimol experiment (monkey e) and the ML recordings (monkey d) were done in only one monkey.

We have performed the muscimol experiment in a second monkey and included it in the revised manuscript. Since didn't see a marked difference between the results for the two subjects, the results are presented as combined. The data for each animal is shown above in Reviewer Fig. 8. As already noted above, the ML experiments were from two monkeys (monkey A and monkey D).

The PR recordings were pooled across two animals, but Monkey B only contributed 14 neurons, responses. For the screening set on Monkey B (PR) it is very difficult to see whether there is a difference in average firing rate between familiar and unfamiliar faces. For the 1000 monkey face set, Monkey B (PR) shows a very large difference with cells barely responding to unfamiliar faces. Interestingly the changes in preferred axis between familiar and unfamiliar faces are less robust (Extended Figure 4). Unfortunately, no behavioral data were collected on Monkey B to see if the behavior explains the smaller differences observed. Considering the variability, I think it would be best not to pool neurons across animals.

We again note that the variability of Monkey B (PR) was likely largely due to the fact that we could only record 14 cells for Experiment 2 before this animal died. We have now recorded more cells (N = 46 cells) in PR in a new animal (Monkey E) and confirmed that results for PR are consistent (Extended Data Fig. 4).

We decided to pool across animals for our main figures and show data for individual animals in an extended data figure because we fear the manuscript will be unreadable otherwise.

Many figures don't have error bars. Figure 2a and Extended Figure 10 (no error bars on the familiar and unfamiliar data points). Figure 2f,g, having error bars on mean neural distance between faces would

give important context to the magnitude of the difference. In extended Figure 4, it would be good to show the similarity matrices for each monkey and area.

Thank you for these suggestions, we have added error bars on the familiar and unfamiliar data points on Fig. 2a and related Extended Data Figures. We also added error bars as shaded area in Fig. 4a, c (previous Fig. 2f, h), please note that some of them are too small to be seen or are barely visible. Similarity matrices for each monkey and area have been added to Extended Data Fig. 4.

We thank the reviewer again for their extremely careful reading of our manuscript and incisive comments and suggestions, which have greatly improved our manuscript.

Reviewer Reports on the First Revision:

Referees' comments:

Referee #1 (Remarks to the Author):

This is the most exemplary response to reviews I have ever seen. All of my concerns have been directly addressed. And, the revised manuscript adds a great deal of importance by showing data from TP that undercut claims made by Landi et al. (Science 2021). I think the overall result is a landmark report that will influence future research and teaching, including my own.

---Ed Connor

Referee #2 (Remarks to the Author):

The authors addressed my comments.

The novelty of the approach and findings is now slightly compromised by the publication of a related study (Landi et al., 2021).

Referee #3 (Remarks to the Author):

While no one can argue with the amount of work that went into this manuscript including the revision, I disagree with the authors' claim that their story is simple. I agree that their hypothesis is simple: face memory is encoded through a long-latency change in axis. However, while they present some support for this hypothesis in face patches AM and PR that paints a simple story; if you zoom out and look at all the recordings in aggregate across the different face patches the story very quickly stops being simple.

Before delving into the complications, the revised manuscript is conceptually weakened by the removal of the claim that the increase in neural distance between familiar faces is related to axis change. That was an interesting result and had the potential of providing a hook to quantitatively compare the physiology to the behavior in the context of the axis change hypothesis.

The authors give two reasons for removing the claim: 1)The result wasn't robust to the way the axis was computed (STA vs linear regression). 2) New recordings from a new area, TP, show an increase in neural distance with familiarity but show no changes in axis.

The first reason validates concerns raised by all three reviewers about the robustness of the axis change. In the revision, the authors include a new analysis revealing that axis change, unlike repetition suppression, is robust across experimental contexts. I am just puzzled by why that analysis is relegated to Extended figure 19, since a main point of the paper is that axis change is more robust

than repetition suppression. I suspect that part of the reason is the absence of a statistical method or framework to evaluate the robustness of the axis change. Extended figure 19 is a good start, but how are we supposed to interpret it, what do we make of the TP results? Are they robust even though they don't show an axis change, or are they less robust because they don't show an axis change? This paper isn't the first challenge to the repetition suppression hypothesis of memory in IT. Meyers and Rust (2018) showed that weighted linear readout of IT responses was a better predictor of visual memory judgements than a strict instantiation of the repetition hypothesis. How is an axis change better? Or is an axis change consistent with the Meyers and Rust results?

While the authors address many of the reviewers' concerns about robustness with suitable analyses, many of the analyses are only in the rebuttal and not in the revised manuscript. Cross-validation analyses addressing overfitting are important and should be included in the manuscript. The analysis that shows that axis change is robust to cell inclusion criteria (Reviewer Figure 2) is also important, and should be done not only for areas AM and PR but also for area TP. Same thing for the PCA analyses.

The addition of recordings with a set of perfectly feature-matched unfamiliar faces adds to the robustness of the result. But I believe the contrast control is still the strongest one. The authors reassuringly confirm in the rebuttal that changing the contrast to 50% has a marked effect on firing rate. It would be good to quantify the firing rate change due to contrast and relate it to the small axis change. Since contrast effects are typically monotonic, it might be even possible to extrapolate to lower contrasts for which faces are still visible. The premise of the contrast control is that contrast affects firing rate but not the axis change, characterizing the limits of that robustness will strengthen the result. To be clearly explained, the contrast control deserves more than a sentence and a subpanel of a figure.

The second reason the authors give for removing the claim that large neural distance is related to axis change is based on new data in area/face patch TP. The new data in TP and reanalysis of data from ML complicate the simple results in areas AM and PR. TP is different from areas AM and PR in that it shows an increase in neural distance with familiarity but little change in axis. Area ML is different from all the other face-patches studied in that it shows no increase in neural distance with familiarity but it shows a change in axis albeit slightly delayed. The authors try to attribute the ML result to feedback, but it is not clear to me why the axis change would be fed back but not the change in neural distance. There might be a way to weave these four results into one simple story, I am just not seeing it in this revised manuscript.

The behavioral data are a good addition. Unfortunately, there is no attempt to quantitatively link the behavioral data to the increase in neural distance for the familiar faces in comparison to the unfamiliar faces. Furthermore since the neural distance is no longer linked to the axis change I am lost as to the significance of the axis change vis-a-vis the behavior.

I understand the instinct to record from TP and address the findings of Landi et al. (2021), unfortunately the TP recordings ended up complicating the results of the current manuscript. To address Landi et al., The authors introduce new experiments which include a new class of faces (faces of other animals). The authors argue that the results with the animal faces contradict the

claims of Landi et al. (2021). This is presented in Extended Figure 5. The results from TP are already hard to reconcile with other results in this paper, trying to reconcile them with other studies is a tall order for one Extended Figure. Even for just TP, the story isn't simple and needs to be explored thoroughly.

To recap, the work is solid and comprehensive. The story is not simple. The short format forces the authors to exclude analyses and shorten description of results and discussion sacrificing clarity. A longer format will give the authors the room to fully describe all their results, include all the different robustness analyses that they performed, and adequately compare their results to previous literature.

As far as impact, the axis change remains a potential code for multiplexing face memory in "sensory" face patches. More work is needed before it is established as "the" code for face memory. Specifically, axis change results across areas need to be reconciled into a consistent if complex story. Furthermore, a quantitative link to behavior is needed.

Minor points:

The figures are small and have a lot packed in them but I noticed some inconsistencies in the data from TP that need to be clarified.

In figure 1g, f TP looks similar to PR in terms of the effects of familiarity. In figure 3a there is a clear distinction between the 36 familiar faces and the 1000 unfamiliar faces but the difference is smaller than PR.

In fig 1g, TP seems to have a higher baseline firing. Responses never seem to go down to 0. In fig 3a, TP normalized firing rate is odd in that there is no prevalence of "orange" or "red" high positive responses for familiar faces like for PR or AM. Is that because of the high baseline firing rate which affects the normalization? How does spontaneous firing/normalization affect the axis changes? TP data from figures 3 a,b and e don't look consistent. There doesn't seem to be much red for familiar faces in A, yet the average time courses figure 3b show a clear difference between familiar and unfamiliar. The histograms in figure 3e on the other hand seem to show the fewest cells that show a difference between familiar and unfamiliar faces. Yet decoding and neural distance for familiar faces are high. Maybe only a few cells are driving the whole effect. How does that impact the axis change?

For area ML and AM pooling across animals makes sense because their results are quite consistent. For area PR and TP less so. PR is consistent across animals when it comes to cosine similarity but the RDMs are wildly different. TP is inconsistent for both. It is consistent in terms of an absence of difference in cosine similarity for familiar versus unfamiliar. But the cosine similarity distributions are shifted from each other.

Graphically, all the histograms are very hard to read. Especially when one histogram is hidden behind another. One solution is to interleave/interdigitate them. The scattergrams in the rebuttal are much clearer and could be a helpful amendment to some of the histograms.

Author Rebuttals to First Revision: Responses to Reviewer Comments

We thank all three reviewers for their careful consideration of our paper.

Referee #1 (Remarks to the Author):

This is the most exemplary response to reviews I have ever seen. All of my concerns have been directly addressed. And, the revised manuscript adds a great deal of importance by showing data from TP that undercut claims made by Landi et al. (Science 2021). I think the overall result is a landmark report that will influence future research and teaching, including my own.

---Ed Connor

We are deeply grateful for these kind words and glad to hear that we have addressed Dr. Connor's concerns.

Referee #2 (Remarks to the Author):

The authors addressed my comments.

The novelty of the approach and findings is now slightly compromised by the publication of a related study (Landi et al., 2021).

We are glad to hear that we have addressed the reviewer's concerns. We think our findings are of even greater interest given the publication of Landi et al., 2021. This paper claimed to find a new brain area, temporal pole face patch TP, that only responds to familiar faces. We show that this claim is false (evidence summarized below), and further show that TP, unlike ML, AM, and PR (Fig. 2b-d), does not show long latency axis change to familiar faces. This last finding underscores the special role of AM and PR in long term memory, in agreement with substantial independent evidence (e.g., see Miyashita, NRN 2019).

As we show in Extended Data Fig. 5, we could not replicate the central claim of Landi et al., 2021:

- Our new recordings reveal that cells in TP show a weaker response on average to familiar faces (Fig. 1c), just like cells in patches AM and PR.
- Our new recordings further reveal that cells in TP respond even more strongly to faces of other species, never seen before, than to human or monkey faces (Extended Data Fig. 5f).
- In our experiments, we explicitly ensured that unfamiliar faces were truly unfamiliar, as they were presented only once per recording site; in contrast, in Landi et al., the same set of "unfamiliar" stimuli were presented across multiple sessions.
- Furthermore, when we re-analyzed the data from Landi et al. ourselves, we found that most of the purported selectivity for familiar faces in TP was present only in a tiny number of cells (Extended Data Fig. 5a), and their main finding depended on using a normalization procedure that over-represented these cells (Extended Data Fig. 5b).

Referee #3 (Remarks to the Author):

While no one can argue with the amount of work that went into this manuscript including the revision, I disagree with the authors' claim that their story is simple. I agree that their hypothesis is simple: face memory is encoded through a long-latency change in axis.

We thank the reviewer for their extremely careful rereading of the manuscript and incisive comments which have greatly improved the manuscript. We are grateful to the reviewer for

acknowledging that our hypothesis—namely, face memory is encoded through a long-latency change in axis—is simple.

However, while they present some support for this hypothesis in face patches AM and PR that paints a simple story; if you zoom out and look at all the recordings in aggregate across the different face patches the story very quickly stops being simple. Before delving into the complications, the revised manuscript is conceptually weakened by the removal of the claim that the increase in neural distance between familiar faces is related to axis change. That was an interesting result and had the potential of providing a hook to quantitatively compare the physiology to the behavior in the context of the axis change hypothesis.

We respectfully disagree that the manuscript is conceptually weakened by removal of the claim that the increase in neural distance between familiar faces is related to axis change. The central point of our paper is that long term memory is accompanied by axis change, *a fundamentally new concept for how memory alters the neural population code*. Our results suggest that, in contrast, the classic concept of neural distance is not a good correlate of long-term memory: In response to the screening stimuli (Experiment 1, in which familiar and unfamiliar stimuli were presented with a 34/16 ratio), neural distance between familiar faces was decreased compared to unfamiliar faces, since the firing rate was reduced. The reduction was not observed in Experiment 2, in which the frequency of presentation of unfamiliar stimuli was significantly higher. Yet long-latency axis change remained apparent in both experiments, and the change was consistent as demonstrated by the significant correlation between the response selectivity for familiar faces between Experiments 1 and 2 (Extended Data Fig. 8e). Thus we think *neural distance is not a robust correlate of long-term memory, unlike axis change*.

In retrospect, we can see why the Reviewer found the story confusing, since we emphasized the finding of increased neural distance for familiar faces in Fig. 4a-d in the previous version of the manuscript. We have now moved these panels to Extended Data Figures related to increased neural distance to Extended Data Fig. 9j, k. Furthermore, we have added the following sentence to the Discussion:

“We speculate that these relative response amplitudes, and associated neural distances and decoding accuracies (Extended Data Fig. 9j, k), may reflect momentary stimulus saliency rather than face memory.”

The authors give two reasons for removing the claim: 1) The result wasn't robust to the way the axis was computed (STA vs linear regression). 2) New recordings from a new area, TP, show an increase in neural distance with familiarity but show no changes in axis.

The first reason validates concerns raised by all three reviewers about the robustness of the axis change.

We respectfully disagree. All axis change results reported in the manuscript were robust. First, we explicitly confirmed axis change to be consistent independent of STA vs. linear regression (Extended Data Fig. 8a, bottom). Second, we found that when familiar and unfamiliar faces were matched in number (36), familiar axes did as well in explaining responses to familiar faces as unfamiliar axes did in explaining responses to unfamiliar faces (Extended Data Fig. 3e, f). Third, we took extreme care to ensure that axis change was not driven by feature differences (Extended Data Fig. 8a, top).

In the revision, the authors include a new analysis revealing that axis change, unlike repetition suppression, is robust across experimental contexts. I am just puzzled by why that analysis is relegated to Extended figure 19, since a main point of the paper is that axis change is more robust than repetition suppression. I suspect that part of the reason is the absence of a statistical method

or framework to evaluate the robustness of the axis change. Extended figure 19 is a good start, but how are we supposed to interpret it, what do we make of the TP results? Are they robust even though they don't show an axis change, or are they less robust because they don't show an axis change?

We agree that context robustness of axis change is an important result. However, we think this analysis is essentially a *control analysis* to confirm robustness, and so it is best placed in Extended Data (Extended Data Fig. 8e) for interested readers. We address TP results in detail further below.

This paper isn't the first challenge to the repetition suppression hypothesis of memory in IT. Meyers and Rust (2018) showed that weighted linear readout of IT responses was a better predictor of visual memory judgements than a strict instantiation of the repetition hypothesis. How is an axis change better? Or is an axis change consistent with the Meyers and Rust results? The Meyer and Rust (2018) proposal of weighted linear readout is NOT a challenge to repetition suppression, but rather a natural extension of repetition suppression to population coding, simply acknowledging that not all neurons are equally suppressed. To contrast their proposal with repetition suppression, the authors introduced the "strict instantiation of the repetition hypothesis", assuming that repetition affects all neurons to the same extent, a straw-man notion at odds with the core principles of repetition suppression. Their primary conclusion is simply that a subset of neurons were suppressed, and weighted linear readout assigning higher weights to that subset of neurons describes data better.

Weighted linear readout would not be able to explain the complete change of sign in response magnitude to familiar versus unfamiliar faces across our two experiments—in particular, the increased response to familiar faces in Experiment 2. Furthermore, long latency axis change is a dynamic code completely different from the type of static neural code of weighted linear read-out. Finally, long latency code change is theoretically supported as a mechanism to remove memory capacity constraints of disentangled representations (Boyle et al., 2023).

While the authors address many of the reviewers' concerns about robustness with suitable analyses, many of the analyses are only in the rebuttal and not in the revised manuscript. Cross-validation analyses addressing overfitting are important and should be included in the manuscript. The analysis that shows that axis change is robust to cell inclusion criteria (Reviewer Figure 2) is also important, and should be done not only for areas AM and PR but also for area TP. Same thing for the PCA analyses.

The cross-validation analysis addressing overfitting (Reviewer Figure 1) is included in the manuscript as Extended Data Fig. 3g, h, and this is referenced explicitly in the main text:

"We found that when familiar and unfamiliar faces were matched in number (36), familiar axes did as well in explaining responses to familiar faces as unfamiliar axes did in explaining responses to unfamiliar faces (Extended Data Fig. 3g, h)."

We thank the Reviewer for the suggestion to include the analysis showing that axis change is robust to cell inclusion criteria (former Reviewer Figure 2). We have now added this to Extended Data Fig. 8a, middle, and also performed the same analysis for area TP (which showed consistent results). We assume by PCA analysis the Reviewer is referring to the former Reviewer Figure 5. Since this analysis was complicated and tangential to the main story, we decided not to include it in the revision (also due to space limitations).

The addition of recordings with a set of perfectly feature-matched unfamiliar faces adds to the robustness of the result. But I believe the contrast control is still the strongest one. The authors reassuringly confirm in the rebuttal that changing the contrast to 50% has a marked effect on firing rate. It would be good to quantify the firing rate change due to contrast and relate it to the small axis change. Since contrast effects are typically monotonic, it might be even possible to

extrapolate to lower contrasts for which faces are still visible. The premise of the contrast control is that contrast affects firing rate but not the axis change, characterizing the limits of that robustness will strengthen the result. To be clearly explained, the contrast control deserves more than a sentence and a subpanel of a figure.

The suggested analysis is indeed interesting, even though extrapolation is sensitive to noise and thus not a completely convincing way of estimating the values outside of the measured range. We did this analysis (Reviewer Fig. 1a) and find that even extrapolating the firing rate to zero (-100%), the expected axis change is significantly smaller than the axis change caused by familiar faces. Moreover, even this small axis change is overestimated, because lower firing rate is correlated with higher trial-trial variation (Reviewer Fig. 1b), resulting in low axis estimation accuracy. This result is indeed informative and *supports our claim that axis change cannot be replicated by firing rate changes caused by change in contrast*. However, we think it is too complicated to interpret and thus we think it is better to not incorporate in the revision.

Reviewer figure 1. Extrapolating effects of contrast on axis change. (a) Axis change vs. percent firing rate change caused by contrast change. Each gray dot is one cell. Axis change was measured by difference of cosine similarity. (b) Trial-trial variability, measured as coefficient of variation at the decreased contrast, as a function of % firing rate change.

The second reason the authors give for removing the claim that large neural distance is related to axis change is based on new data in area/face patch TP. The new data in TP and reanalysis of data from ML complicate the simple results in areas AM and PR. TP is different from areas AM and PR in that it shows an increase in neural distance with familiarity but little change in axis. Area ML is different from all the other face-patches studied in that it shows no increase in neural distance with familiarity but it shows a change in axis albeit slightly delayed. The authors try to attribute the ML result to feedback, but it is not clear to me why the axis change would be fed back but not the change in neural distance. There might be a way to weave these four results into one simple story, I am just not seeing it in this revised manuscript.

We think all of these results do cohere in a simple story, namely, that long latency axis change is the correlate of face memory. As explained above, we think that neural distance is not a good correlate of face memory; this conclusion mainly comes from the huge, qualitative difference in neural distance between Experiments 1 and 2.

Re ML, we did not intend to suggest that axis change is fed back from more anterior patches. What we had written was:

"The population average firing rate [in ML] showed a sustained divergence between responses to familiar and unfamiliar faces much later than in AM (160 ms compared to 140 ms in AM, Extended Data Fig. 9c), suggesting that ML may receive a familiarity-specific feedback signal from AM."

We simply meant here that the difference in the mean response to familiar and unfamiliar faces might have been fed back from AM, not axis change. Axis change is likely accomplished by local recurrent dynamics within ML (since it occurred *earlier* in ML than in AM and PR).

The behavioral data are a good addition. Unfortunately, there is no attempt to quantitatively link the behavioral data to the increase in neural distance for the familiar faces in comparison to the unfamiliar faces. Furthermore since the neural distance is no longer linked to the axis change I am lost as to the significance of the axis change vis-a-vis the behavior.

Indeed, we do not claim that axis change improves neural discriminability. As we explained above, we have removed the sections on increased neural distance to Extended Data to make absolutely clear that we do not think neural distance is the correlate of face memory.

Our new behavioral test (now Extended Data Fig. 3a, b) primarily shows that monkeys do perceive familiar faces differently from unfamiliar faces, better distinguishing them under conditions of blur. We agree it does not directly link to axis change.

We have some ideas for how to link axis change to face memory capacity that would be exciting to pursue in the longer term: First, we could expose subjects to a substantial number of faces for a long time to saturate their memory capacity. Then, to make a causal link to axis change, we could apply neural perturbation techniques that specifically disrupt long latency responses (e.g., using holographic optogenetics with new ultra-fast opsins (Adesnik & Abdeladim, 2021)). Even short of this, we think it may be possible to find a correlation if a relatively large population of animals were tested, since there should be innate variations in memory capacity among individuals.

I understand the instinct to record from TP and address the findings of Landi et al. (2021), unfortunately the TP recordings ended up complicating the results of the current manuscript. To address Landi et al., The authors introduce new experiments which include a new class of faces (faces of other animals). The authors argue that the results with the animal faces contradict the claims of Landi et al. (2021). This is presented in Extended Figure 5. The results from TP are already hard to reconcile with other results in this paper, trying to reconcile them with other studies is a tall order for one Extended Figure. Even for just TP, the story isn't simple and needs to be explored thoroughly.

We respectfully disagree that the story from TP is not simple. The original claim by Landi et al. was that TP is a memory center for familiar faces, responding exclusively to familiar faces. As we describe in detail in response to Reviewer 2 above, we not only show that this claim is contradicted by data from our own new experiments, but we also show, by reanalyzing the data from Landi et al., that their conclusions were based on a very small number of cells and an unconventional normalization procedure.

With respect to the relationship of our TP results to our central claim, we think they add support to our central claim by:

- 1) providing an additional control showing that the observation of axis change in AM and PR is not a stimulus-related artifact (since cells in TP did not show long-latency axis change, unlike in AM and PR (Fig. 2b-d),

2) showing that long-latency axis change is a circumscribed phenomenon that can be localized to regions of the temporal lobe implicated in long-term memory by substantial independent evidence (e.g., see Miyashita, NRN 2019).

To recap, the work is solid and comprehensive. The story is not simple. The short format forces the authors to exclude analyses and shorten description of results and discussion sacrificing clarity. A longer format will give the authors the room to fully describe all their results, include all the different robustness analyses that they performed, and adequately compare their results to previous literature. As far as impact, the axis change remains a potential code for multiplexing face memory in “sensory” face patches. More work is needed before it is established as “the” code for face memory. Specifically, axis change results across areas need to be reconciled into a consistent if complex story. Furthermore, a quantitative link to behavior is needed.

As explained above, we think all our results strongly point to long latency axis change as the correlate of long-term face memory. We are very grateful to the Reviewer for pointing out the dissonance generated by our distracting extended discussion of neural distance change in the former Fig. 4a-d. As already mentioned, we have now moved these figure panels to Extended Data Fig. 9j, k. We have also added the following clarifying sentence to the Discussion:

“We speculate that these relative response amplitudes, and associated neural distances and decoding accuracies (Extended Data Fig. 9j, k), may reflect momentary stimulus saliency rather than face memory.”

We grant that a causal experiment perturbing the long latency axis code and demonstrating an effect on memory behavior would be the gold standard for demonstrating that axis change is indeed the code for long term memory. Technological limitations make this dream experiment very difficult at present. (For perspective, we note that the causal role of grid cells in spatial navigation remains contentious to this day, e.g., see Ginosar et al., Neuron 2023, a decade after their discovery was awarded the Nobel Prize). We have re-titled the paper, “Temporal multiplexing of perception and memory codes in IT cortex” to reflect that our advance is more modest than conclusively establishing the code for long term memory.

Minor points:

The figures are small and have a lot packed in them but I noticed some inconsistencies in the data from TP that need to be clarified.

In figure 1g, f TP looks similar to PR in terms of the effects of familiarity. In figure 3a there is a clear distinction between the 36 familiar faces and the 1000 unfamiliar faces but the difference is smaller than PR.

The apparent discrepancy is because the difference in familiarity effect between PR and TP is not in the average responses across familiar or unfamiliar faces, but in the distribution of responses to individual faces. Fig. 1g shows responses averaged across familiar or unfamiliar faces; directly comparable to Fig. 1g is Fig. 3b, showing similar familiarity effects on average for PR and TP. In contrast, Fig. 3a shows responses to individual face images; directly comparable to Fig. 3a is Fig. 1c (monkey faces), showing responses to individual faces: both show a smaller familiarity effect in TP. In short, the results for TP are consistent between Fig. 1 and Fig. 3.

In fig 1g, TP seems to have a higher baseline firing. Responses never seem to go down to 0. In fig 3a, TP normalized firing rate is odd in that there is no prevalence of “orange” or “red” high positive responses for familiar faces like for PR or AM. Is that because of the high baseline firing rate which affects the normalization? How does spontaneous firing/normalization affect the axis changes?

First, we use the original neural responses, firing rate, to compute the axis for each neuron, not normalized responses. The axis is a direction in face feature space that only depends on the relative responses across different faces; in principle, it is not affected by a constant offset (spontaneous firing). We only use normalization when necessary (e.g., when visualizing average neural responses across the population, showing firing rate biases the result for high firing rate neurons).

Second, the lack of orange for familiar faces in Fig. 3a TP is not due to the baseline firing rate. Comparing AM vs. PR (Fig. 1g, left), the differences in baseline firing rate are much larger, but they look similar in Figure 3a. Instead, the lack of orange for familiar faces in Fig. 3a TP is because the proportion of high firing rate responses to unfamiliar faces is similar to familiar ones in TP, while lower in PR (Reviewer Fig. 2, left). The normalization scales the maximum face response to one, thus we don't see the prevalence of orange for familiar faces in TP. However, this higher response to unfamiliar faces does not affect much the mean responses to unfamiliar faces (Reviewer Fig. 2, right), because the proportion of low firing responses is also higher (Reviewer Fig. 2, left), i.e., the variance in the responses to unfamiliar faces was higher in TP.

Reviewer Figure 2. Distribution of neuronal responses (baseline firing rate subtracted). Left, distribution of responses by pooling together all cells and faces. Right, distribution of responses averaged across all familiar or unfamiliar faces for each cell.

TP data from figures 3 a,b and e don't look consistent. There doesn't seem to be much red for familiar faces in A, yet the average time courses figure 3b show a clear difference between familiar and unfamiliar. The histograms in figure 3e on the other hand seem to show the fewest cells that show a difference between familiar and unfamiliar faces. Yet decoding and neural distance for familiar faces are high. Maybe only a few cells are driving the whole effect. How does that impact the axis change?

As explained above, the apparent discrepancy between Fig. 3a, b is due to the slightly higher variance of responses to unfamiliar faces in TP, but not the mean responses. And this also explains why a lower proportion of cells in TP could pass the statistical test for difference between

mean response to familiar vs. unfamiliar (Fig. 3e, dark bar). Consistent with this, the decoding performance in TP is indeed affected (Fig. 3c), but the neural distance for face centroids (Fig. 3d) is not, as it only depends on the mean responses, not the response distribution.

To check if the familiarity effect in TP was driven by a few cells, we analyzed the familiarity decoding performance as the function of cell number (Reviewer Fig. 3), and it showed that the decoding performance increased almost linearly with number of cells in all areas, so the effect is not due to a few cells.

In any case, the results of Fig. 3a-e are orthogonal to axis change. Specifically, differences in mean response to familiar and unfamiliar faces does not impact axis change. The axis depends only on the response pattern to different faces, e.g., we could arbitrarily scale the responses to familiar or unfamiliar faces to create whatever distribution we want for Fig. 3e, but this would not change the preferred axis.

Reviewer Figure 3. Accuracy for decoding familiarity as a function of number of cells. Average responses between 50-300 ms were used.

For area ML and AM pooling across animals makes sense because their results are quite consistent. For area PR and TP less so. PR is consistent across animals when it comes to cosine similarity but the RDMs are wildly different. TP is inconsistent for both. It is consistent in terms of an absence of difference in cosine similarity for familiar versus unfamiliar. But the cosine similarity distributions are shifted from each other.

We don't think there is inconsistency in PR or TP data related to our central claim (long latency axis change). The RDM from monkey E for PR is weaker because we were only able to collect 14 cells. We have now added the number of cells that were obtained for each individual animal to the figure legend for Extended Data Fig. 4 to make this clear. Our main point regarding TP is that there is an absence of difference in cosine similarity for familiar versus unfamiliar, which as the Reviewer states, is clear in both animals. We think the main figures would be unreadable if we were to show all data for each animal individually. At the same time, to ensure transparency, we show all data for individual monkeys in Extended Data Fig. 4, so interested readers may consult these panels.

Graphically, all the histograms are very hard to read. Especially when one histogram is hidden behind another. One solution is to interleave/interdigitate them. The scattergrams in the rebuttal are much clearer and could be a helpful amendment to some of the histograms.

We are unclear why the Reviewer finds the histograms hard to read. The histograms (e.g., Fig. 2b) use transparency, and the three tones (blue, orange, overlap) seem clear to us.

References

Boyle, L. M., Posani, L., Irfan, S., Siegelbaum, S. A., & Fusi, S. (2023). Tuned geometries of hippocampal representations meet the demands of social memory. In *bioRxiv* (p. 2022.01.24.477361). <https://doi.org/10.1101/2022.01.24.477361>